# Clinical practice guideline adherence in oncology: A qualitative study of insights from clinicians in Australia

**Mia Bierbaum**[1]*, **Frances Rapport**[1], **Gaston Arnolda**[1,2], **Geoff P. Delaney**[2,3], **Winston Liauw**[2,4], **Ian Olver**[5], **Jeffrey Braithwaite**[1,2]

**1** Australian Institute of Health Innovation, Macquarie University, Sydney, Australia, **2** Centre for Research Excellence in Implementation Science in Oncology, Sydney, Australia, **3** SWSLHD Cancer Services, Liverpool, Australia, **4** SESLHD Cancer Service, Kogarah, Australia, **5** School of Psychology, University of Adelaide, Adelaide, Australia

* mia.bierbaum@hdr.mq.edu.au

**Data Availability Statement:** All relevant data are within the manuscript and the tables. Additional qualitative data cannot be made public because it describes participant experiences which are not de-

## Abstract

### Background

The burden of cancer is large in Australia, and rates of cancer Clinical Practice Guideline (CPG) adherence is suboptimal across various cancers.

### Methods

The objective of this study is to characterise clinician-perceived barriers and facilitators to cancer CPG adherence in Australia. Semi-structured interviews were conducted to collect data from 33 oncology-focused clinicians (surgeons, radiation oncologists, medical oncologists and haematologists). Clinicians were recruited in 2019 and 2020 through purposive and snowball sampling from 7 hospitals across Sydney, Australia, and interviewed either face-to-face in hospitals or by phone. Audio recordings were transcribed verbatim, and qualitative thematic analysis of the interview data was undertaken. Human research ethics committee approval and governance approval was granted (2019/ETH11722, #52019568810127).

### Results

Five broad themes and subthemes of key barriers and facilitators to cancer treatment CPG adherence were identified: Theme 1: CPG content; Theme 2: Individual clinician and patient factors; Theme 3: Access to, awareness of and availability of CPGs; Theme 4: Organisational and cultural factors; and Theme 5: Development and implementation factors. The most frequently reported barriers to adherence were CPGs not catering for patient complexities, being slow to be updated, patient treatment preferences, geographical challenges for patients who travel large distances to access cancer services and limited funding of CPG recommended drugs. The most frequently reported facilitators to adherence were easy accessibility, peer review, multidisciplinary engagement or MDT attendance, and transparent CPG development by trusted, multidisciplinary experts. CPGs provide a reassuring

identifiable and consent for release was not provided by study participants. The South West Sydney Local Health District Human Research Ethics Committee (HREC) provides oversight for the data collected, and by policy requires any use of this data to be directly approved by the HREC. For these reasons, data may only be made available upon request made to the Corresponding Author and the South West Sydney Local Health District HREC. Please direct data requests to the following non-author email at: SWSLHD-Ethics@health.nsw.gov.au.

**Funding:** MB is supported by an Australian government Research Training Program Scholarship stipend associated with the Australian Institute of Health Innovation, Macquarie University (ID:9100002). The funders had no role in study design, data collection and analysis, decision to publish, or preparation of the manuscript. https://www.dese.gov.au/research-block-grants/research-training-program.

**Competing interests:** The authors have declared that no competing interests exist.

**Abbreviations:** CPG, Clinical Practice Guidelines; COREQ, Consolidated Criteria for Reporting Qualitative Studies checklist; GP, General Practitioner; GRT, Guideline Recommended Treatment; H, Haematologists; i, interview number; LHD, Local Health District; MO, Medical Oncologists; NSLHD, North Sydney LHD; NSW, New South Wales, Australia; PBS, Pharmaceutical Benefits Scheme; RO, Radiation Oncologists; S, Surgeons; SES LHD, South-Eastern Sydney LHD; SWS LHD, South-Western Sydney; TA, Thematic Analys; TGA, Therapeutic Goods Administration; WSLHD, Western Sydney LHD; VMOs, Visiting Medical Officers.

framework for clinicians to check their treatment plans against. Clinicians want cancer CPGs to be frequently updated utilising a wiki-like process, and easily accessible online via a comprehensive database, coordinated by a well-trusted development body.

## Conclusion

Future implementation strategies of cancer CPGs in Australia should be tailored to consider these context-specific barriers and facilitators, taking into account both the content of CPGs and the communication of that content. The establishment of a centralised, comprehensive, online database, with living wiki-style cancer CPGs, coordinated by a well-funded development body, along with incorporation of recommendations into point-of-care decision support would potentially address many of the issues identified.

## Background

The burden of cancer is large in Australia, with the number of new cases (excluding non-melanoma skin cancer) estimated to reach 150,782 in 2021[1] (population of 26 million people[2]). Clinical Practice Guidelines (CPGs) are designed to support clinical decision-making, based on the best evidence, reduce unwarranted clinical variation [3], minimise healthcare expenditure and improve care [4], however non-adherence to CPGs may be justifiable in various circumstances. Emerging literature in Australia indicates that cancer CPG adherence is associated with improved patient outcomes, resulting in increased survival rates [5,6]. However, across the Australian health system, less than 60% of care has been estimated to be adherent to CPGs [7].

Sub-optimal rates of adherence to cancer Guideline-Recommended Treatment (GRT) specifically, have been identified in Australia, across a variety of cancer streams [8,9]. For example, GRT was received by: just over half of the patients with cervical cancer in NSW (2005–2011) [10]; two-thirds of patients in SA (2000–2010) with stage C colon cancer, and nearly half of stage B and C rectal cancer patients [5]; two-thirds of selected patients in NSW (2006–2011) with Non-Small Cell Lung Cancer (NSCLC) [11,12]; and only one-third of patients in NSW with melanoma (2006–2007) [13]. Rates of GRT have been found to be underutilised across a variety of cancers in Australia [8] and internationally [14–21].

### Factors that enhance CPG adherence

Dissemination strategies that enhance adherence to CPGs include: face-to-face [22–25] and web-based educational workshops [26], educational outreach programs [27–29], printed materials [22], computerised reminders [22,27] (particularly point-of-care decision support) [30,31], and support by local opinion leaders [32], particularly when used in combination [27]. Adapting CPGs to local contexts can also improve the acceptability of CPGs for the user [3]. Modern dissemination of CPGs has shifted to electronic formats [26], with CPGs now available on multiple platforms, including hand-held devices, wiki-based CPGs [33], and electronic decision-tools at the point-of-care [26]. Compared to printed formats, electronic formats potentially increase accessibility, enabling quicker updates with feedback, while nudging clinicians towards adhering to GRTs, such as appropriate antibiotic use and hand hygiene [34].

## Clinician attitudes towards cancer CPGs

A recent systematic review [35] identified that globally, clinicians are generally positive about cancer-specific CPGs, however negative attitudes, and barriers to adherence, persist. Key barriers include concerns about: recency of evidence, cookbook medicine, the need to account for patient complexities, weak evidence, and side-effects associated with GRT, as well as patient treatment preferences, poor accessibility to CPGs, ingrained clinical practice habits and concerns that GRT will increase costs of healthcare [35].

The review also identified key factors that facilitated adherence to GRT including: adapting CPGs to local needs, endorsement from medical colleges and colleagues, educational sessions, MDT meetings, and access to the recommended medicines, as well as clinician agreement with CPGs [35]. CPGs were considered useful, convenient sources of information, and educational tools that support treatment decision making, assist clinicians in litigation issues, and were generally perceived to enhance patient care [35].

Effective Implementation of CPGs needs to take into account local requirements and characteristics of the health system [36]. Considering the range of adherence rates in Australia, it is important to develop our understanding of clinician attitudes towards cancer CPGs. The aim of this study was to examine in-depth, clinician attitudes towards and perceived barriers and facilitators to cancer CPG adherence, to inform implementation strategies for cancer CPGs in the future [37].

## Methods

This manuscript reports the findings from an inductive, exploratory qualitative study, and conforms with the S1 Checklist [38]. This study was informed by the interdisciplinary framework developed by Gurses et al [39]. It encompasses the qualitative component of a multiphase sequential mixed-methods study [37,40]. The findings from these interviews will inform the quantitative data collection in the proceeding phase.

### Ethics statement

Human research ethics committee approval was attained granted by the South Western Sydney Local Health District Human Research Ethics Committee, and Macquarie University Human Research Ethics Committee (2019/ETH11722, #52019568810127), as well as governance approval at each hospital site.

### Recruitment and data collection

Clinicians who met the eligibility criteria (Box 1) and worked in one of 7 major hospitals offering cancer services across South-Western Sydney Local Health District (SWSLHD), South-Eastern Sydney LHD (SESLHD), Western Sydney LHD (WSLHD), and North Sydney LHD

---

### Box 1. Eligibility criteria

| To be eligible, participants needed to meet all four criteria: |
| --- |
| 1) They were Radiation Oncologists (ROs), Medical Oncologists (MOs), Surgeons, Haematologists, or Registrars in any of these discipline areas; and |
| 2) They currently treated patients with a cancer diagnosis, in Australia; and |
| 3) They were willing to provide written informed consent and to participate in the study; and |
| 4) They were willing and able to complete the interview in English. |

(NSLHD) were invited to participate in an interview; these four LHDs contain approximately half of the population of New South Wales (NSW), Australia [41]. As more than a quarter of all cancer services are located in NSW [42], these clinicians are expected to be representative of Australian cancer clinicians. Purposive sampling [43,44] was used to recruit interview participants, with promotional emails sent to targeted clinicians by key hospital contacts; snowball sampling was conducted with interviewees invited to pass study invitations to colleagues [43]. This approach ensured a targeted sample of clinicians from specific disciplines and of varying seniority, but preserved clinician autonomy by allowing self-selection to guide interview participation, in line with Australian ethical guidelines for research [45]. Each person who contacted the research team was sent a participant information sheet and consent form (PICF), which included details about the study aim and design, the qualifications of the interviewer and contact details for the study team. The PICF was signed before an interview commenced. Interviews were conducted between October 2019-January 2020.

The interview topic guide was piloted in interviews with three clinicians. Following analysis of pilot data, no amendments were made (see S1 Appendix). The remaining semi-structured interviews were conducted, and data from all interviews were included in the full analysis. No repeat interviews were conducted.

Interviews were conducted either over-the-phone or face-to-face in hospital, by the lead researcher (MB, BSc, MPH, PhD candidate), an experienced female qualitative researcher. The interviewer had no prior relationship with any interviewee. Interviews were approximately 30 minutes in duration. All participants were offered a gift voucher as a token of appreciation for their participation. All interviews were audio recorded, transcribed verbatim, and deidentified.

## Data analysis

An iterative, inductive thematic analysis (TA) [46,47] approach was used to obtain insight into the experiences and perceptions of participating clinicians regarding adherence to cancer CPGs. This allowed patterns to emerge from the data, through re-reading and coding of the transcripts (by MB), establishing a deeper understanding of the data (Steps 1–3 of TA: Data familiarisation, generation of initial codes and theme searching) [48]. Themes were refined (Step 4 of TA: Theme review) [48], while reflexively examining the influence of the authors' assumptions on data analysis [47], and acknowledging anticipated themes informed by a topical systematic review [35]. Recruitment and data analysis ceased once thematic saturation occurred and no new codes were identified [43]. Analysis was conducted using NVIVO version 12.4.0 [49]. The resulting coding framework was discussed during development (with FR), with iterative adjustments made to the themes and codes following discussion (Step 5 of TA: Theme definition and naming) [48]. This two coder technique enabled the corroboration of the thematic framework and for team consensus to be reached on the coding terminology [44]. The final framework was validated by FR who read and coded 5 interviews to ensure trustworthiness and methodological rigor [50] (see S2 Appendix). All remaining transcripts were then recoded (by MB) using the finalised thematic framework [44]. The frequency with which codes were identified across the interview transcripts, was calculated in order to identify how many clinicians raised each subtheme, giving an indication of whether attitudinal trends existed across disciplines [51].

'Member checking' was employed to enhance data credibility and minimise potential misinterpretation of data [44]. Following completion of thematic data analysis, a summary of the preliminary findings was sent to each participant, providing them with an opportunity to verify, reject or clarify researcher thematic interpretation of findings. Checking-back was considered important to minimise the potential for misinterpretation. Any clinician feedback would be returned to the study team for consideration and integrated into the final findings.

## Results

### Demographics

Thirty-three interviews were completed, including 3 pilot interviews. Most clinicians were aged 40–49 years (33.3%), practiced in SWSLHD (54.5%), and were staff specialists (75.8%). Breast cancer (30.3%) and Haematological cancers (30.3%) were the most common cancers the clinicians worked with. Half of the clinicians (51.5%) reported working in only one cancer stream and nearly half of the clinicians had commenced specialist practice within the preceding decade (2010–2019) (48.5%) (Table 1).

**Table 1. Demographic data of interview participants.**

| Clinician characteristics | | n (33) | % |
|---|---|---|---|
| **Age** | 30-39y | 9 | 27.3% |
| | 40-49y | 11 | 33.3% |
| | 50-59y | 7 | 21.2% |
| | 60+y | 6 | 18.2% |
| **LHDs where each clinician predominantly practices** | SWSLHD | 18 | 54.5% |
| | WSLHD | 8 | 24.3% |
| | NSLHD | 5 | 15.2% |
| | SESLHD | 2 | 6.1% |
| **Discipline of clinician** | Radiation oncology | 10 | 30.3% |
| | Medical oncology | 9 | 27.3% |
| | Surgery | 8 | 24.2% |
| | Haematology | 6 | 18.2% |
| **Predominant cancer streams clinicians practice in (more than one per clinician)** | Breast cancer | 10 | 30.3% |
| | Haematological cancer | 8 | 24.2% |
| | Lung cancer/thoracic cancer | 8 | 24.2% |
| | Melanoma/ skin cancer | 7 | 21.2% |
| | Gastrointestinal cancers | 7 | 21.2% |
| | Genitourinary cancers | 4 | 12.1% |
| | Sarcoma | 4 | 12.1% |
| | Thyroid/ Endocrine cancer | 3 | 9.1% |
| | Gynaecological cancer | 2 | 6.1% |
| | Head and Neck cancer | 2 | 6.1% |
| | Other (General, Paediatric surgical oncology, Abdopelvic) | 3 | 9.1% |
| **Professional position of clinician** | Staff specialist | 25 | 75.8% |
| | Visiting Medical Officers (VMOs) | 6 | 18.2% |
| | Fellow | 1 | 3.0% |
| | Registrar | 1 | 3.0% |
| **Year of graduation as a specialist in oncology** | 2015–2019 | 9 | 27.3% |
| | 2010–2014 | 7 | 21.2% |
| | 2000–2009 | 7 | 21.2% |
| | 1990–1999 | 6 | 18.2% |
| | 1980–1989 | 4 | 12.1% |

*South Western Sydney LHD (SWSLHD), Western Sydney HD (WSLHD), Northern Sydney LHD (NSLHD), South Eastern Sydney LHD (SESLHD), Sydney LHD (SLHD); Gastrointestinal cancers (including Oesophageal, stomach, biliary system, small intestine, large intestine, colon, rectum, anus, pancreatic, liver cancers), Genitourinary cancers (including prostate, kidney, bladder and testicular cancers and cancers of the penis), Haematological cancers (including Leukaemia/ Lymphoma).

### Response rate and member checking

Invitations were sent to 66 clinicians to participate in an interview, and an unknown number by snowballing; 35 clinicians contacted the study team, and of those, 33 clinicians completed the interview. Five clinicians responded to the invitation for member checking and provided confirmatory feedback. This limited feedback was positive and did not substantively change interpretation (5/33). The characteristics of the non-respondents are unknown.

### Themes that emerged from the interviews

Five key themes with subthemes were identified during analysis of the interviews: CPG content; Individual clinician and patient factors; Access to, awareness of and availability of CPGs; Organisational and cultural factors; and CPG development and implementation factors (see S2 Appendix). Barriers and facilitators to CPG adherence were identified within each theme, and the proportion of clinicians who contributed to each of the subthemes are presented, according to their medical discipline (see Table 2). Clinicians were assigned a label based on their sequential interview number. Quotes representing each theme and subtheme are presented in Table 3.

## Theme 1: CPG content

### Subtheme 1.1: Applicability of recommendations to patient population

**Barriers.**   CPGs not catering for patient complexities such as comorbidities, performance status, age, or the ability to tolerate treatment, was a barrier to CPG adherence raised by many clinicians. When CPGs were not applicable to patients, clinicians made clinical judgements and modified CPG recommendations, tailoring treatment to individual needs, referred to as *the art of medicine*. It was unclear from the interviews whether these modifications would be considered warranted variation within the scope of the CPG or considered non-adherent.

A third of clinicians reported modifying CPG recommendations when concerned that treatments would not be well tolerated by patients, or when patients were perceived to be able to tolerate more aggressive treatment than the CPG recommends. Such modifications are not necessarily non-adherent, as some CPGs include recommendations for modifications for certain patient groups. When these modifications are made, they are often justified and approved through peer review or in MDT meetings, recorded in electronic patient records, and in letters back to General Practitioners (GPs) and patients. One common justification for modifications, was that the evidence underpinning CPGs was gathered from clinical trials comprised of patient cohorts who are generally healthier and younger than patients being seen by clinicians, reducing the applicability of the CPGs.

**Facilitators.**   Locally adapted or Australian CPGs provide context-specific information and were seen to be more likely adhered to. CPGs reflective of peer-accepted practice were considered useful as were CPGs that provide options to modify recommendations for specific patient populations.

Half of the clinicians commented that CPG adherence was a good measure of quality of care, indicating where practice variation lies, and possible reasons for variation, so long as the guideline was up-to-date, noting that a lack of adherence must be interpreted carefully. Many clinicians found that CPGs create a coherent framework within which to discuss patients, and this is particularly useful for decision making around complex cases, for unfamiliar clinical scenarios, less common cancers, or for new treatments. CPGs also provide reassurance for junior clinicians, and for busy clinicians working with a cancer with which they are less familiar.

**Table 2. The frequency of clinicians reporting each theme and subtheme.**

| | Codes (interview number) | Total (33) | MO (9) | RO (10) | S (8) | H (6) |
|---|---|---|---|---|---|---|
| **Theme 1: CPG content** | | | | | | |
| **Subtheme 1.1: Applicability of recommendations to patient population** (i1, 2, 3, 5, 6, 7, 8, 9, 10, 11, 12, 14, 15, 16, 17, 18, 19, 20, 21, 22, 23, 24, 25, 26, 27, 28, 29, 30, 31, 32, 33) | | **31** | 9 | 10 | 7 | 5 |
| **Barriers** | CPGs do not, or cannot cater for all patient complexities (i2, 3, 5, 6, 8, 11, 12, 14, 15, 16, 18, 19, 20, 21, 22, 23, 24, 26, 27, 28, 29, 30, 31, 32, 33) | **25** | 6 | 10 | 6 | 3 |
| | Modifications are often made due to concerns that CPG recommendations would not be well tolerated by patients, or would lead to unnecessary side effects, or adverse events (i1, 3, 7, 8, 9, 14, 16, 18, 25, 26, 29, 31), or when patients are perceived to be able to tolerate more aggressive treatment than the CPG recommends (i21) | **13** | 5 | 4 | 2 | 2 |
| | Modifications are justified and approved through peer review or in MDT meetings, recorded in electronic patient records, and in letters back to GPs and patients (i6, 9, 10, 12, 15, 18, 20, 25, 27, 29, 30, 32) | **12** | 4 | 4 | 2 | 2 |
| | CPGs underpinned by evidence from clinical trial cohorts that are not representative of the patient population (i3, 6, 9, 10, 12, 15, 20, 21, 25, 27) | **10** | 5 | 3 | - | 2 |
| | The art of medicine/oncology/clinical practice means clinicians often make modifications to CPG recommendations (i5, 6, 14, 17, 30) | **5** | 2 | - | 3 | - |
| | CPGs that are not multidisciplinary in their approach (i31, i33) | **2** | - | 1 | 1 | - |
| | CPG timeframes that are unrealistic (i20, i26) | **2** | 1 | - | 1 | - |
| **Facilitators** | CPGs provide a reassuring framework for clinicians to check their treatment plans against (i2, 3, 6, 8, 9, 12, 13, 14, 15, 16, 17, 18, 19, 20, 21, 22, 24, 27, 28, 29, 30, 31, 32, 33) | **24** | 6 | 9 | 7 | 2 |
| | CPG adherence is a good measure of quality of care, as it indicates where variation lies, and possible reasons for variation, so long as the guideline is up-to-date, and lack of adherence is interpreted carefully (i1, 2, 3, 4, 5, 8, 11, 12, 13, 16, 17, 19, 24, 25, 28, 30) | **16** | 4 | 4 | 4 | 4 |
| | CPGs provide assurance for junior clinicians (i1, 3, 4, 5, 7, 8, 10, 11, 14, 15, 17, 19, 28, 33) | **14** | 6 | 2 | 3 | 3 |
| | The framework provided by CPGs is considered useful for decision making during complex cases, for unfamiliar clinical scenarios, less common cancers, or new treatments (i1, 3, 4, 5, 7, 13, 14, 16, 17, 18, 30, 32, 33) | **13** | 4 | 1 | 6 | 2 |
| | CPGs help clinicians reach consensus when there is debate over the sequence of treatment from different disciplines (i7, 8, 9, 12, 14, 18, 21, 23, 28, 29, 31) | **11** | 3 | 7 | 1 | - |
| | Existence of locally adapted CPGs facilitates adherence (i5, 11, 19, 21, 22, 23, 25, 27, 29, 31) | **10** | 1 | 6 | - | 3 |
| | CPGs help clinicians to reach consensus in borderline cases or when the evidence base is controversial (i5, 6, 8, 9, 12, 14, 17, 18, 31) | **9** | 4 | 3 | 2 | - |
| | CPGs were generally seen as helpful, educational tools, particularly for common cancer cases (i1, 3, 4, 5, 10, 13, 14, 19, 22) | **9** | 3 | 1 | 2 | 3 |
| | CPGS are perceived to reduce clinical variation (i1, 2, 3, 8, 9, 23, 27) and improve patient care (i10) | **8** | 4 | 3 | - | 1 |
| | CPG recommendations that are reflective of peer accepted practice, particularly for common cancers (i3, 11, 17, 23, 27, 30, 33) | **7** | 1 | 2 | 3 | 1 |
| | CPGs that provide options to modify recommendations (i10, 16, 18, 25) | **4** | - | 1 | - | 3 |
| | CPGs provide assurance for busy clinicians working with a cancer they are not an expert in (i2, 7, 17, 18) | **4** | 1 | 2 | 1 | - |
| | CPGs that provide information about specific dose information, organs at risk and patient side effect profiles (i2, 6) | **2** | 1 | 1 | - | - |
| | CPGs provide assurance when treatments and evidence are changing rapidly (i1) | **1** | 1 | - | - | - |
| **Subtheme 1.2: Degree of evidence and level of agreement with evidence underpinning CPGs** (i1, 2, 3, 4, 5, 6, 7, 8, 9, 10, 11, 12, 13, 14, 15, 16, 17, 18, 20, 21, 22, 23, 24, 25, 26, 27, 29, 30, 32, 33) | | **Total (30)** | **MO (9)** | **RO (7)** | **S (8)** | **H (6)** |
| **Barriers** | CPGs underpinned by rapidly changing evidence (i1, 3, 4, 5, 6, 8, 9, 10, 11, 12, 13, 16, 17, 20, 23, 24, 29, 30, 33) | **19** | 7 | 3 | 5 | 4 |
| | CPGs underpinned by poor or emerging evidence base (i2, 4, 7, 9, 12, 14, 15, 16, 21, 22, 24, 26, 27, 29, 32) | **15** | 3 | 5 | 4 | 3 |
| | When there is a lack of evidence underpinning CPGs (i2, 4, 7, 12, 14, 15, 17, 18, 22, 27, 29), or recommendations are expert consensus-based (i17, 27, 29), clinicians often prefer to rely on their own clinical judgement when making treatment decisions | **11** | 2 | 5 | 2 | 2 |
| | A lack of agreement with the interpretation of evidence underpinning the CPG, particularly when the evidence is controversial (i4, 12, 14, 26, 29) or when CPGs vary in recommendations, it can be difficult to decide which guideline to follow (i21) | **6** | - | 3 | 2 | 1 |
| | Good patient survival outcomes lead to practice variation and lower adherence (i2, 4, 17) | **3** | - | 1 | 1 | 1 |
| | Clinicians are reluctant to change practice in line with CPG recommendations, without first critically appraising the evidence underpinning the practice change (i2,7) | **2** | 1 | 1 | - | - |

(*Continued*)

**Table 2.** (Continued)

| | Codes (interview number) | Total (33) | MO (9) | RO (10) | S (8) | H (6) |
|---|---|---|---|---|---|---|
| Facilitators | CPG recommendations underpinned by high quality and clear, uncontroversial evidence (i1, 3, 5, 7, 10, 11, 12, 17, 18, 22, 27, 30) | 12 | 4 | 3 | 2 | 3 |
| | Consensus-based CPGs were considered better than no CPG being available (i7, 16) | 2 | 1 | - | - | 1 |
| | CPG recommendations that have been shown to increase survival of patients (i17) | 1 | - | - | 1 | - |
| | When multiple CPGs are similar in content, with little variation (i32) | 1 | - | - | 1 | - |
| | When there are multiple CPGs to choose from, to tailor treatments to specific patient contexts (i1) | 1 | 1 | - | - | - |
| **Subtheme 1.3: Format- ease of use, references to evidence, and inclusion of patient resources** (i1, 2, 3, 4, 5, 6, 7, 8, 9, 10, 11, 12, 16, 17, 18, 19, 20, 21, 22, 26, 27, 28, 29, 30, 31, 32) | | **Total (26)** | **MO (8)** | **RO (9)** | **S (4)** | **H (5)** |
| Barriers | CPGs that do not include background references or justification for the recommendations, and do not explicitly state whether recommendation are based on evidence or expert opinion (i4, 6, 7, 28) | 4 | 2 | 1 | - | 1 |
| | CPGs that are difficult to navigate (i2, 9, 20) | 3 | 2 | 1 | - | - |
| | CPGs that are not complex or informative enough (i1, 2) | 2 | 1 | 1 | - | - |
| | Patient resource section of CPGs are often not useable if not available in languages other than English (i20) | 1 | 1 | - | - | - |
| Facilitators | Provision of a concise summary of evidence that includes justifications and reference to the clinical trials underpinning recommendations (i1, 2, 3, 5, 6, 7, 9, 10, 18, 19, 22, 27, 28, 30, 32) | 15 | 6 | 5 | 2 | 2 |
| | Good lay out, easy to read and user friendly (i1, 6, 10, 11, 17, 18, 21, 27, 28, 32) | 10 | 2 | 4 | 2 | 2 |
| | Provision of schedule and dose information provides assurance for clinicians that they are practicing appropriately and accurately (i2, 6, 7, 8, 9, 20, 28, 31), especially if they work across multiple cancer streams (i20) | 8 | 5 | 3 | - | - |
| | Inclusion of patient resources within a CPG can help to increase treatment decision transparency when discussing treatment plans with patients (i2, 6, 8, 12, 20, 26, 31) | 7 | 3 | 3 | 1 | - |
| | CPGs that highlight what level of evidence the recommendations are based on, whether the evidence is controversial, or the recommendations consensus based (i5, 7, 18, 19, 30) | 5 | 2 | 2 | 1 | - |
| | Inclusion of information on side effects for clinicians to reference when making treatment decisions and monitoring patients (i1, 3, 6, 8, 16) | 5 | 4 | - | - | 1 |
| | Comprehensive, and informative CPGs that include multiple treatment options (i1, 2, 16, 21) | 4 | 1 | 2 | - | 1 |
| | Inclusion of a decision tree or flow chart (i2, 32) | 2 | - | 1 | 1 | - |
| **Subtheme 1.4: How up-to-date CPGs are** (i1, 2, 3, 4, 5, 6, 7, 8, 9, 10, 11, 12, 13, 14, 15, 16, 17, 18, 19, 20, 21, 22, 23, 24, 25, 26, 27, 28, 29, 30, 31, 32, 33) | | **Total (33)** | **MO (9)** | **RO (10)** | **S (8)** | **H (6)** |
| Barriers | CPGs that are slow to be updated (i1, 2, 3, 4, 5, 6, 8, 10, 11, 12, 13, 14, 16, 20, 23, 24, 25, 26, 27, 29, 30, 31, 33), or have a lack of standardised updating procedures, and not knowing when the next version will be released (i20) | 23 | 6 | 6 | 6 | 5 |
| | Clinicians not receiving notifications regarding CPG updates (i12, 13, 18, 20, 21, 25, 29, 30, 31, 32) | 10 | 1 | 5 | 3 | 1 |
| | Outdated CPGs are often require local protocols to be developed (i2, 21), for example, to guide contouring | 2 | - | 2 | - | - |
| Facilitators | CPGs being updated regularly (i1, 3, 7, 9, 13, 15, 16, 17, 20, 23, 25, 27, 28, 29, 31, 32) | 16 | 6 | 5 | 3 | 2 |
| | Clinicians receiving notifications about updates to CPGs from colleges, colleagues or CPG developers (i13, 16, 18, 20, 21, 22) | 6 | 1 | 2 | 1 | 2 |
| **Subtheme 1.5: Prescriptiveness of CPG recommendations** (i1, 2, 4, 5, 6, 7, 8, 11, 12, 13, 14, 16, 19, 20, 21, 26, 33) | | **Total (17)** | **MO (6)** | **RO (4)** | **S (4)** | **H (3)** |
| Barriers | CPG content that is too broad, and not detailed enough for complex cases (i2, 4, 5, 6, 7, 11, 12, 13, 14, 21, 33) | 11 | 3 | 3 | 3 | 2 |
| | CPG content that is too rigid, not taking account of emerging evidence (i1, 8, 12, 19, 26) | 5 | 2 | 2 | 1 | - |
| | Less treatment clarity for second- and third-line treatment due to a lack of evidence, leads to more practice variation (i1, 11, 12, 13) | 4 | 1 | 1 | 1 | 1 |
| | Inclusion of conservative recommendations (i2, 12, 21) | 3 | - | 3 | - | - |
| **Theme 2: Individual clinician and patient factors** | | | | | | |
| **Subtheme 2.1: Clinician personality, and the impact of CPGs on autonomy** (i1, 3, 5, 6, 7, 10, 12, 13, 14, 15, 16, 17, 18, 19, 22, 23, 24, 26, 27, 28, 29, 30, 32, 33) | | **Total (24)** | **MO (6)** | **RO (7)** | **S (8)** | **H (3)** |

(*Continued*)

**Table 2.** (Continued)

| | | Codes (interview number) | Total (33) | MO (9) | RO (10) | S (8) | H (6) |
|---|---|---|---|---|---|---|---|
| **Barriers** | | CPGs are not specific rules or directives about treatments that clinicians should strictly adhere to (i1, 3, 6, 7, 10, 12, 14, 15, 16, 17, 18, 19, 22, 24, 27, 28, 29, 30, 32, 33) | 20 | 5 | 6 | 6 | 3 |
| | | Clinician hubris, with some strong personalities influencing treatment decisions in MDTs, potentially acts as a barrier to adherence (i6, 14, 19, 20, 21, 23, 26, 28, 31, 32) | 10 | 2 | 5 | 3 | - |
| | | As specialists, some clinicians felt they no longer needed to refer to CPGs (i3, 7, 12, 14, 22, 27, 28, 29, 30) | 9 | 2 | 4 | 2 | 1 |
| | | Individual clinical equipoise challenges the ability of clinicians to accept changing treatment options (i19, 21, 23, 28, 30, 32) | 6 | - | 4 | 2 | - |
| | | Some clinicians are perceived to dislike having treatments dictated to them by CPGs and CPG developers (i10, 17, 23, 26) | 4 | - | 1 | 2 | 1 |
| | | Concern that CPGs can lead to cookbook, or "cookie cutter" medicine, reducing clinician autonomy (i23, 26) | 2 | - | 1 | 1 | - |
| | | Clinician concern that CPG adherence can lead to under-dosing (i7, 15) | 2 | 2 | - | - | - |
| **Facilitators** | | CPGs increase junior clinician autonomy, as it provides them with an independent mechanism to confirm treatment plans (i3, 5, 16, 19) | 4 | 2 | 1 | - | 1 |
| | | CPGs help clinicians overcome clinical equipoise, and provide guidance, to reduce clinical variation (i12, 28). | 2 | - | 2 | - | - |
| | | CPGs allow clinician freedom to choose treatments, not limiting professional autonomy (i13) | 1 | - | - | 1 | - |
| **Subtheme 2.2: Generational and disciplinary differences in perceptions towards CPGs** (i1, 3, 6, 7, 8, 10, 14, 15, 16, 19, 23, 26, 27, 28, 31) | | | Total (15) | MO (6) | RO (5) | S (2) | H (2) |
| **Barriers** | | Senior clinicians are less inclined to refer to CPGs, compared to more junior clinicians (i1, 3, 8, 19, 23, 26, 31) | 7 | 3 | 3 | 1 | - |
| | | Some clinicians are biased by a preference for their own discipline, or financially incentivised by fee-for-service, to complete treatment with the patient rather than engage in multidisciplinary care (i7, 10, 14, 15, 27, 29, 31), in urological care in particular (i27, 31) | 7 | 2 | 3 | 1 | 1 |
| | | Junior clinicians' practice can be influenced by the preferences of senior clinicians (i6, 16, 27, 28) | 4 | 1 | 2 | - | 1 |
| **Subtheme 2.3: Litigation concerns** (i5, 6, 7, 8, 9, 10, 11, 12, 14, 15, 16, 17, 18, 19, 20, 24, 25, 28, 30, 32, 33) | | | Total (21) | MO (7) | RO (4) | S (6) | H (4) |
| **Barriers** | | Following guidelines blindly, due to apprehension about litigation related to non-adherence, could lead to patients not receiving the best practice (i7, 11, 19, 25, 33) | 5 | 1 | 1 | 1 | 2 |
| | | Concerns around litigation may be a reason CPGs are not developed, particularly regarding treatment doses (i10). | 1 | - | - | - | 1 |
| **Facilitators** | | Possible litigation and the need to justify and communicate treatment decisions clearly, and demonstrate that clinicians are practicing according to the evidence (i5, 6, 7, 8, 9, 11, 12, 14, 15, 16, 17, 18, 20, 24, 28, 30, 32, 33) | 18 | 7 | 3 | 6 | 2 |
| **Subtheme 2.4: Patient age, comorbidities, preferences and logistics** (i1, 2, 3, 4, 5, 7, 8, 10, 11, 12, 13, 14, 15, 16, 18, 19, 20, 21, 22, 25, 26, 27, 28, 30, 31, 32, 33) | | | Total (27) | MO (7) | RO (8) | S (6) | H (6) |
| **Barriers** | | Patient preference (i1, 2, 3,4, 5, 8, 10, 11, 12, 13, 14, 18, 19, 20, 21, 26, 27, 28, 31, 32, 33), and concern about side effects (i1), toxicity, and treatment tolerability (i2, 12), with some patients rejecting certain treatments based on anecdotal experience of friends and family receiving particular treatments (i10, 12, 18, 21) | 21 | 5 | 8 | 5 | 3 |
| | | Clinician concern about patients' older age (i2, 3, 14, 15, 16, 18, 20, 21, 22, 27), frailty (i18, 22, 27), fitness (i20), performance status (i14,15), comorbidities (i2, 3, 13, 14, 15, 21, 30), contraindications (i25), organ impairment (i16) | 13 | 3 | 4 | 3 | 3 |
| | | Geographic challenges and logistics for rural and remote patients travelling long distances to access treatments (i3, 4, 12, 18, 20, 22, 26, 28, 30, 33) | 10 | 2 | 3 | 3 | 2 |
| | | Clinician concern about toxicity or potential side effects of a treatment (i1, 2, 3, 16, 31), concern about the psychosocial impacts of treatments (i28), and the impact of patient treatment history in terms of treatment tolerability (e.g., the impact of past radiation on current radiation treatment plans) (i21, 28) | 7 | 2 | 4 | - | 1 |
| | | Concern that adhering to a CPG recommendation will lead to poorer patient outcomes (i7) | 1 | 1 | - | - | - |
| | | Limited patient access to family and peer support (i13). | 1 | - | - | 1 | - |
| **Theme 3: Access to, awareness of and availability of CPGs** | | | | | | | |
| **Subtheme 3.1: Access to, awareness of and availability of CPGs** (i2, 3, 4, 5 6, 7, 8, 9, 10, 11, 12, 13, 15, 16, 17, 18, 19, 20, 21, 22, 23, 25, 27, 28, 29, 30, 31, 32) | | | Total (28) | MO (8) | RO (10) | S (4) | H (6) |
| **Barriers** | | Hard to access CPGs (i2, 6, 7, 8), published in a journal that is not open access (i25), requiring a login or membership to access the guideline (as passwords are often forgotten) (i2, 6, 9, 20, 21, 23, 28) | 10 | 5 | 4 | - | 1 |
| | | Not many CPGs are available for rare cancers (i2, 4, 7, 9, 17, 18, 22, 25) | 8 | 2 | 2 | 1 | 3 |
| | | Not many (if any) local Australian CPGs are available in specific fields (i4, 6, 7, 11, 22, 29) | 6 | 2 | 1 | - | 3 |
| | | International CPGs not applicable locally (i4, 10, 15, 23, 27) | 5 | 1 | 2 | - | 2 |
| | | Other clinicians' limited awareness of CPGs (i5, 19, 28, 29) or limited knowledge of where to access them (i16, 19) | 5 | 1 | 3 | - | 1 |
| | | Poor Wi-Fi infrastructure in hospitals can limit real time access to CPGs while on wards (i20, 31), and hospital internet site restrictions can prevent clinicians from accessing external CPG specific sites (i6) | 3 | 2 | 1 | - | - |
| | | Other clinicians who are not up-to-date with the literature in general, are perceived as less likely to adhere to CPGs (i31) | 1 | - | 1 | - | - |

(*Continued*)

**Table 2.** (Continued)

| | Codes (interview number) | Total (33) | MO (9) | RO (10) | S (8) | H (6) |
|---|---|---|---|---|---|---|
| Facilitators | Easy access to guidelines (i2, 3, 7, 9, 11, 12, 13, 15, 16, 17, 18, 19, 20, 21, 22, 23, 25, 28, 30) | 19 | 5 | 7 | 3 | 4 |
| | Electronic CPGs (i15, 16, 17, 23, 25, 28, 30) available via an app are easier to use (i13, 15, 20, 28) | 9 | 2 | 2 | 3 | 2 |
| | Local CPGs, produced in hospitals or departments (i2, 10, 11, 16, 19, 21, 25), and Australian CPGs (i20, 30) are preferred | 9 | 1 | 3 | 1 | 4 |
| | CPGs published in open access journals (i18, 23, 25), or peer reviewed, reputable journals (i2, 25) | 4 | - | 3 | - | 1 |
| | Preferred international CPGs, particularly as international CPGs tend to be more updated (i11, 13, 15, 23) | 4 | 1 | 1 | 1 | 1 |
| | CPG websites that require no password (i20, 23, 25, 32) | 4 | 1 | 1 | 1 | 1 |
| | CPGs that are free to download (i18, 23, 25) | 3 | 2 | - | - | 1 |
| | CPGs are a good mechanism to keep clinicians up-to-date with the literature (i17) | 1 | - | - | 1 | - |
| **Theme 4: Organisational and cultural factors** | | | | | | |
| **Subtheme 4.1: Access to treatments recommended by CPGs, resource availability and clinician time** (i1, 2, 3, 4, 5, 6, 7, 8, 9, 10, 11, 12, 15, 16, 17, 19, 20, 22, 23, 24, 25, 26, 27, 28, 29, 31, 32, 33). | | Total (28) | MO (9) | RO (8) | S (5) | H (6) |
| Barriers | A shortage of or limited availability of CPG recommended drugs (i3, 6, 12, 22, 32), including international CPG recommended drugs not TGA approved, or PBS funded in Australia (i1, 2, 3, 4, 5, 6, 7, 8, 10, 11, 15, 20, 22, 23, 25, 29, 33). | 19 | 8 | 4 | 2 | 5 |
| | High clinician workload (i19), limited staffing (i28), a lack of clinician time (i16, 20, 23, 31) and a lack of cancer care coordinators (i24) can prevent clinicians from looking up CPG recommendations, as it can be quicker to ask a colleague for advice (i20) | 7 | 1 | 4 | 1 | 1 |
| | Having limited access to resources (i27) including treatment and technology (i5, 12, 28, 33), for example specific radiotherapy machines, can lead to clinicians using a different technique appropriate to the technology they have access to (i19) | 6 | 1 | 4 | 1 | - |
| | Cost of international CPG adherent treatments, if the treatments are not publicly funded in Australia (i3, 4, 15, 31) | 4 | 2 | 1 | - | 1 |
| Facilitators | Regular meetings to discuss CPGs, protocols, and practices (i1, 8, 9, 17, 19, 22, 24, 25, 29, 31), and purposeful hospital provision of protected clinician time to read, discuss and contribute to CPGs and the literature (i10, 23, 31) | 12 | 3 | 4 | 2 | 3 |
| | Organisational support and the provision of adequate resources (i2, 24, 31, 33), the availability of care coordination for scans and treatment (i20, 24), as well as the infrastructure and use of flexible treatment plans to provide home-based treatment (i20, 26) | 6 | 1 | 2 | 3 | - |
| | When there is no PBS funding for a specific CPG recommended drug, some access schemes by pharmaceutical companies or Local Health Districts, can enable patients to access those drugs (i10, 15, 22, 33) | 4 | 1 | 1 | - | 2 |
| | CPGs save clinicians time by concisely summarising the evidence, so long as they are up-to-date (i27, 28, 32) | 3 | - | 2 | 1 | - |
| | CPGs also support clinician advocacy for more resources to be publicly available (i33) | 1 | - | - | 1 | - |
| **Subtheme 4.2: A culture of peer review or multidisciplinary review of treatment plans** (i1, 2, 5, 6, 7, 8, 9, 10, 11, 12, 13, 14, 15, 16, 17, 18, 19, 20, 21, 22, 23, 24, 25, 26, 27, 28, 29, 30, 31, 32, 33) | | Total (31) | MO (8) | RO (10) | S (8) | H (5) |
| Barriers | Limited access to peer review or multidisciplinary review of treatment plans for private practicing clinicians (i10, 11, 14, 15, 17, 18, 20, 21 23, 25, 26, 31, 32) and rural/regional clinicians (i10, 17, 18, 19, 20, 23, 27) | 15 | 2 | 6 | 4 | 3 |
| | Poor MDT attendance, or poor multidisciplinary engagement (i14, 18, 19, 20, 23, 24, 27, 29, 30), or poor relationships in MDTs (i32) | 10 | 1 | 5 | 4 | - |
| | Peer review occurs less frequently for common cancers or straight forward cases (i11, 31, 32) | 3 | - | 1 | 1 | 1 |
| | Peer review is limited for rare cancers with fewer specialists in the field (i17, 19) | 2 | - | 1 | 1 | - |
| | Hospital culture or preference for more aggressive or less aggressive treatment than what is prescribed by the CPG recommendations (i7, 14) | 2 | 1 | - | 1 | - |
| | Lack of quality imaging to support the MDT treatment review process (i14) | 1 | - | - | 1 | - |
| | Limited interaction with other clinicians and therefore, limited exposure to new treatment strategies (i13) | 1 | - | - | 1 | - |
| Facilitators | Multidisciplinary engagement or MDT attendance (i5, 6, 8, 9, 10, 12, 13, 14, 15, 17, 18, 19, 20, 21, 23, 24, 25, 26, 27, 28, 30, 31, 32, 33) | 24 | 6 | 8 | 8 | 2 |
| | Peer review of treatment decisions (i1, 6, 9, 11, 17, 18, 19, 20, 21, 22, 24, 27, 29, 31, 33) | 15 | 4 | 6 | 3 | 2 |
| | Peer expectation to adhere to CPGs, and fear of looking negligent if non-adherent (i2, 5, 9, 16, 17, 18, 21, 25, 26, 29), and knowing that peers follow specific CPG recommendations (i2) | 10 | 2 | 4 | 2 | 2 |
| | A culture of valuing multidisciplinary care (i15, 20, 32), and a CPG-focused within clinician training (i6, 15, 16) | 5 | 3 | - | 2 | - |
| | Clinical leaders who encourage CPG adherence (i2, 6, 10, 19, 28) | 5 | 1 | 3 | - | 1 |
| | A culture of error reporting (i21) and documenting treatment decisions (i31) | 2 | - | 2 | - | - |
| | Good relationships between multidisciplinary team members, teamwork and timely peer support (i19, 32, 33). | 3 | - | 1 | 2 | - |

*(Continued)*

**Table 2.** (Continued)

| Codes (interview number) | Total (33) | MO (9) | RO (10) | S (8) | H (6) |
|---|---|---|---|---|---|
| **Subtheme 4.3: Referral pathways** (i8, 10, 14, 17, 18, 20, 25, 27, 29, 31) | **Total (10)** | **MO (2)** | **RO (4)** | **S (2)** | **H (2)** |
| Barriers — Patient referral pathways (i8, 10, 14, 18, 25, 27, 29, 31) that circumvent multidisciplinary review | 8 | 1 | 4 | 1 | 2 |
| Lack of awareness by GPs (and patients) of the importance of multidisciplinary review (i10, 17, 18, 20, 29, 31) | 6 | 1 | 3 | 1 | 1 |
| **Theme 5: Development and implementation factors** | | | | | |
| **Subtheme 5.1: Development, adaptations, and review of CPGs, by an expert development committee** (i2, 3, 4, 5, 6, 10, 11, 12, 13, 14, 17, 18, 21, 22, 23, 25, 27, 28, 29, 30, 31, 32, 33) | **Total (23)** | **MO (3)** | **RO (9)** | **S (6)** | **H (5)** |
| Barriers — Limited time is a barrier for clinicians to be involved in CPG development (i23, 31). Development, updating, and maintaining CPGs was seen as a slow and difficult process (i4, 5, 10, 11, 12, 13, 23, 27, 29, 30, 33). | 12 | 1 | 6 | 3 | 3 |
| CPGs that are perceived to be biased, either toward a particular disciplinary based treatment (i14, 17, 29), by clinician agenda (with biased weighting of evidence) (i12, 14, 27, 28, 31, 32, 33), or by pharmaceutical company influence on the committee developing the CPG (i6, 17, 25) | 11 | 1 | 5 | 4 | 1 |
| Facilitators — CPGs developed by trusted and respected experts (i2, 3, 5, 10, 11, 12, 17, 18, 21, 22, 23, 27, 28, 29, 30, 32) in a transparent and methodical way (i10, 12, 27, 30), with multidisciplinary representation on the development committee (i12, 17, 29, 30), as well as patient representatives (i29, 30), to avoid bias | 16 | 2 | 8 | 3 | 3 |
| **Subtheme 5.2: CPG Dissemination and Implementation Strategies** (i1, 2, 3, 4, 5, 8, 9, 10, 11, 12, 16, 18, 19, 21, 22, 23, 25, 26, 27, 28, 29, 30, 31, 32, 33) | **Total (25)** | **MO (5)** | **RO (10)** | **S (4)** | **H (6)** |
| Barriers — Clinical audits of adherence rates do not accurately reflect the reasons for modifying CPG recommendations, or the need to take patient needs into account (i3, 4, 9, 16, 22, 26, 28, 30, 32, 33), highlighting that low CPG adherence may reflect a poorly developed or poor-quality CPG (i2, 9, 33) | 11 | 2 | 2 | 4 | 3 |
| Facilitators — Endorsement of the CPG by trusted organisations such as tumour groups, or authorities who are well known and well published (i2, 5, 10, 12, 18, 21, 23, 25, 28) | 9 | 1 | 6 | - | 2 |
| Clinical audits (i1, 3, 11, 18, 19, 21, 27, 29, 31) | 9 | 2 | 6 | - | 1 |
| Effective dissemination of CPGs through marketing and distribution by the CPG development group (i32), publication of CPGs in high quality journals (i2, 5, 18, 30), and dissemination and discussion regarding CPGs at conferences (i10) | 6 | 1 | 2 | 2 | 1 |
| Education sessions provided by tumour reference groups (i2), and discussions in journal clubs (i5, 19, 29) to increase clinician awareness of CPGs | 4 | 1 | 3 | - | - |
| The incorporation of CPGs into decision tools, such as drop-down treatment options that are pre-programmed into electronic prescribing data record management systems (i8, 9, 21, 29) | 4 | 2 | 2 | - | - |
| **Subtheme 5.3: Suggested development and implementation improvements** (i2, 4, 5, 6, 8, 9, 10, 11, 12, 13, 14, 17, 19, 20, 21, 22, 23, 24, 25, 27, 29, 30, 31, 32, 33) | **Total (25)** | **MO (5)** | **RO (8)** | **S (7)** | **H (5)** |
| Broader clinician input, with wider consultation (i5, 6, 9, 11, 12, 13, 14, 17, 19, 20, 21, 22, 24, 30), with international collaboration to develop CPGs, (i2, 27), and greater opportunities for clinicians to provide feedback regarding the logistics and availability of treatments reflected in the CPG (i5) | 16 | 4 | 3 | 5 | 2 |
| A nationally resourced, centralised, well trusted CPG development body with access to good infrastructure (i29, 31), for quick and efficient CPG development (i2, 5, 6, 10, 11, 25, 27, 29, 30, 32, 33) | 12 | 2 | 3 | 3 | 3 |
| Adapt or tailor international CPGs to local Australian needs (i2, 4, 5, 11, 23, 27, 29, 33) or local, hospital specific CPGs (i21, 29) | 9 | 1 | 5 | 1 | 2 |
| Development of a comprehensive, continuously updated, dynamic, wiki-like CPGs database, managed by a well-resourced national group (i5, 14, 27, 30, 32, 33) | 6 | 1 | 1 | 4 | - |
| The involvement of junior clinicians such as registrars, and trainees in the development process (i9, 10, 19, 20), with authorship enhancing individuals' CVs (i20, 31) | 5 | 2 | 2 | - | 1 |
| Contributions to CPG development could be rewarded through CPG points from the college of physicians, (i9, 10, 11, 20, 31) or financial incentives (i31) | 5 | 2 | 1 | - | 2 |
| If all CPGs were available in a centralised database, then clinicians could sign up to get notifications about updates of specific CPGs (i6, 12, 30, 32) | 4 | 1 | 1 | 2 | - |
| A comprehensive CPG extension to online Australian eviQ protocol resource (i8, 10, 11, 27) | 4 | 1 | 1 | - | 2 |
| CPGs should include treatment sequencing algorithms (like decision trees and flow charts) (i11, 20, 29) | 3 | 1 | 1 | - | 1 |
| CPG development should incorporate real world data for cancers with limited clinical trial evidence (i4, 20, 32) | 3 | 1 | - | 1 | 1 |
| CPGs should include patient resources about their treatment (i8) available in multiple languages and printable (i20) | 2 | 2 | - | - | - |
| CPGs should include links to diet and exercise CPGs and psychosocial care recommendations (i20), links to databases to access information about the clinicians available for consultation prior to treatment (i8). CPGs should also include treatment timeframes that are realistic allowing for imaging and pathology delays (i20). | 2 | 2 | - | - | - |

Note: "I" refers to interview number; N refers to total clinicians; MO refers to Medical Oncologists (i1,3,5,6,7,8,9,15,20); RO refers to Radiation Oncologists (i2, 12, 18, 19, 21,23, 27, 28,29, 31); S refers to Surgeons including one Gynae oncologist (i13, 14, 17, 24, 26, 30, 32, 33); H refers to Haematologists (i4, 10, 11, 16, 22, 25).

CPGs help clinicians reach consensus in borderline cases, particularly when there is debate over the order of treatment modality. They provide a reassuring framework for clinicians to check their treatment plans against, and are generally seen as helpful, educational tools, that reduce clinical variation and improve patient care.

### Subtheme 1.2: Degree of evidence and level of agreement with evidence underpinning CPGs

**Barriers.** Many clinicians discussed how CPG adherence is limited when the evidence base is still emerging. In this case, when recommendations are expert consensus based, clinicians prefer to rely on their own clinical judgement, which may result in low CPG adherence. Similarly, when patient survival outcomes are good, regardless of the treatment provided, higher practice variation results. Adherence is also limited in areas with rapidly changing evidence, especially when emerging evidence indicates better outcomes for patients than current GRT. A lack of agreement with the interpretation of evidence underpinning the CPG was a barrier to adherence, particularly if the evidence is controversial or various CPGs provide different recommendations.

**Facilitators.** Just over a third of clinicians commented that it was important that GRTs were underpinned by high quality, uncontroversial evidence, and that this facilitated adherence.

### Subtheme 1.3: Format, ease of use and references to evidence

**Barriers.** The format of CPGs was considered important, with some CPGs being difficult to navigate if they are too complex. CPGs that do not include references and justifications for recommendations, or explicitly state whether recommendations are based on evidence or expert opinion, were also poorly regarded.

**Facilitators:** Important content factors included CPGs having a good lay out, being easy to read and user friendly, being comprehensive, and including multiple treatment options. Inclusion of information on side effects was considered useful for clinicians to reference when making treatment decisions and monitoring patients.

Provision of schedule and dose information provides assurance for clinicians that they are practicing appropriately and accurately, especially if working across multiple cancer streams. Inclusion of patient resources within the CPG was also an important component for CPGs, as they help to increase treatment decision transparency, and aid communication when discussing treatment plans with patients. Inclusion of concise summaries of evidence with references, that highlight the level of evidence that recommendations are based on, and whether the evidence is controversial or consensus based were highly valued.

### Subtheme 1.4: How up-to-date CPGs are

**Barriers**: Most clinicians noted that one of the main barriers to adherence was that CPGs are often outdated. Many also suggested that not receiving notifications regarding CPG updates was a barrier.

**Facilitators:** CPGs being updated regularly with notifications about CPG updates from colleges, colleagues or CPG developers were considered facilitating factors for adherence.

### Subtheme 1.5: Prescriptiveness of CPG recommendations

**Barriers:** CPG content being too broad and not detailed enough for complex cases, too rigid, not taking account of emerging evidence, or containing conservative recommendations, were

**Table 3. Quotes representing each theme and subtheme.**

| Themes/subthemes | Quotes |
|---|---|
| **Theme 1: CPG content** | |
| SUBTHEME 1.1: Applicability of recommendations to patient population | **Barriers**<br>"The guidelines are written, you've got stage X disease, give treatment Y, but that isn't taking into account the fact that the patient is aged, has five comorbidities, [their] performance status is variable. Variations in practice are often around doctors looking at that evidence on the one side of the coin in the guidelines, and on the other side of the coin, they're in the room and they're trying to apply those guidelines to somebody for whom the evidence doesn't even exist. . . If you look at chemotherapy, the evidence is almost invariably based on persons that have good performance status with minimal comorbidities, normal renal function, normal liver function. And then you've got the reality, which is the patient that's in the bed, who is declining, who has lost more weight than anyone was allowed in the clinical study. His liver function is a bit worse. His kidney function is a bit worse. You know who maybe has mild cognitive impairment, and you've got to work out what do you have to do with them. . .So that's just sort of buyer beware process that so long as you know the parameters upon which a guideline is set then you can make sure that you adjust the decision-making process around those things" (i15, MO)<br>"It's a fine balance..[Treatment A] is a really difficult regiment to tolerate. Patients hate it. We've had some really bad toxicities. But if you strictly follow the guideline, that is what's recommended. But to be fair, you can use any of the second generation, and some guidelines do say, actually you can use any of the second generations. They're probably just as good (i1, MO)<br>"Treating patients is sometimes quite complex and it's not, there's not a blanket rule for everyone, and trial patients which these guidelines stem from, these trial protocols, are looking at a completely different population to the clinician, and that is probably applicable to about 50% of the population of the patients that we treat, in reality" (i3, MO)<br>"Medicine's part science, part art. So particularly, you know, tech surgery for example, 80 percent science, 20 percent art, you know, as an interpretation. So, the guidelines are there to, I guess, minimize the art bit. So, I think [CPGs are] good, but they should be guidelines and not a mantra" (i17, S)<br><br>**Facilitators**<br>"The advantages of [CPGs] are they give you, like I said, confidence in your decisions, added confidence that the decisions that you're making are clinically safe and medico-legally backed up. I guess. For want of a better phrase. I think with like guidelines, they summarise the evidence really well, so they save you time. And are more efficient. And yeah, they can, particularly for myself, as a junior consultant, they give you confidence and they're just a good reference point to guide your practice and to reassure you, often, if the decision you made is what the guidelines say. You know, you go, 'I think I'm going to do this', and you go to the guideline and it agrees with you. It's reassuring" (i28, RO)<br>"I find them useful, and most clinicians find them very useful, in their journey from moving towards being novice, to being experienced. You learn from them, but once you've learned what they have to offer, you [don't] refer back to them. So, it's like an instruction manual for using your TV set. You might flick through it to learn how to use it, but you're not referring back to it when you are using the remote control ever again, once you've figured out the basics" (i33, S)<br>"The only way to reduce variation is adherence to guidelines" (i23, RO) |
| SUBTHEME 1.2: Degree of evidence and level of agreement with evidence underpinning CPGs | **Barriers**<br>"The conference that I went to a few weeks ago, it's not in the guideline now, but it will be in 6 months when they go through all the policy changes. It's too late by then. We need to bring in the data now and say this is what's happening, and this is what's likely to be clinical practice changing. That's how we have to do it, often decisions for the latest treatments are done well before the actual guidelines are formed" (i6, MO)<br>"There is an acceptance on the part of most, I think, most mature oncologists, and oncological surgeons, that the evidence for a lot of the practices that we do, is not high quality. So, even if you do something that isn't 100% evidence based, you're often tailoring patient therapy to their particular situation, and a different treatment paradigm may work. You can have two people with identical tumours, and one treatment paradigm may work with one person and not for another person, depending on their social circumstance, personal circumstances, their age, or patient preference. It's just about being sufficiently flexible. And not sort of being a slave to a protocol" (i32, S)<br>"Some guidelines, they may be well established and evidence-based. So, you would have no problem sticking to those type of guidelines, but there are many conditions where there may not be much literature available or there may not be a randomised controlled study available, and then what happens there? The guidelines may be recommended by the expert opinion on it, and you don't, I mean expert guidelines are one particular individual's opinion, so you don't necessarily need to follow if there is not much evidence for it" (i22, H)<br>"Thyroid cancer is a very benign disease. It is rare as well, it's not common. So as such, there's a lot of variation in practice, but the outcome remains very good. So, you know there's little impetus to actually form guidelines in Australia for that reason because, you know, whether you do a total thyroidectomy or whether you do a . . .central neck dissection, it doesn't really alter the outcome by much" (i17, RO)<br><br>**Facilitators**<br>"It definitely depends on how the guidelines have been devised. If they have been following the literature with a lot of evidence base for those guidelines, I think we would be great to adapt those guidelines. And then other guidelines, there may be some new medications, which have one single study, and then put into the guidelines. Then, you would take with a pinch of salt. Whereas, based on randomised control studies, good evidence to show that particular medication or regimen is better, then you would use that without much hesitancy. So that's the debate we have in the department when we present each disease group to discuss the protocols to adapt into our department" (i22, H) |

*(Continued)*

**Table 3.** (Continued)

| Themes/subthemes | Quotes |
|---|---|
| SUBTHEME 1.3: Format, ease of use and references to evidence, and inclusion of patient resources | **Barriers**<br>"If there's just one paragraph, 'this is what we should use, full stop', and there's nothing else on the background I think 'Oh no', because I'll have to dig it up and I'll be like 'I'm not going to use that again, it is too hard'" (i6, MO)<br><br>**Facilitators**<br>"The ultimate thing that I really want is, if you are making a recommendation, I want to make sure there's sort of a reference to why they came to the conclusion so I can track down the literature" (i6, MO)<br><br>"I think [CPGs are] an excellent, highly reliable guide to consensus interpretation of high-level evidence basically, and they often succinctly…the way they set it out is extremely educational and helpful to a busy practicing clinician, because they'll have a summary there. They actually say, 'Look, here this is the bottom line. And then there's the explanation, what the evidence is, where there's areas of controversy, but the bottom line is we suggest you do blah, blah, blah'" (i5, MO)<br><br>"The two main ones are details about treatment schedules with chemotherapy and doses, a comparison of doses because I'm doing a lot of cancers at the moment. I want to make sure that I'm being as accurate as possible and I, and it'll be when I'm charting chemo and making sure that care plans in our local e-health records and prescribing the doses are concordant with the guidelines on eviQ. And then the second main reason I used [CPGs] in clinic is to print out the patient information" (i20, MO) |
| SUBTHEME 1.4: How up-to-date CPGs are | **Barriers**<br>"[CPGs] never evolve fast enough to take into account new literature" (i33, S)<br><br>**Facilitators**<br>"Being updated. So, most European guidelines, they get updated very quick, almost every six months, and with new trials coming out, they immediately do amendments" (i23, RO) |
| SUBTHEME 1.5: Prescriptiveness of CPG recommendations | **Barriers**<br>"I guess there is little evidence and skin is a varied practice, there are many ways to treat skin cancer. I think guidelines don't offer that kind of detail. We can do different doses and fractionations depending on the patient age and concerns about cosmesis. So, a lot of intricacies, when it comes to treating skin cancer which is a bit, I haven't seen that in guidelines. So, that's usually experiential. That's how I treat skin cancer, and asking colleagues" (i2, RO) |
| **Theme 2: Individual clinician and patient factors** | |
| SUBTHEME 2.1: Clinician personality, and the impact of CPGs on autonomy | **Barriers**<br>"My own opinion about guidelines is, they are a guide, and we can't particularly use them for every single patient. [CPGs are] good in general if you look at a population" (i22, H)<br><br>"I think hubris is a really bad characteristic of some doctors and some surgeons, in particular. People become fixed in their approach, and unwilling to be flexible to modify their approach, based on changing guidelines or changing models of practice, and that can cause problems, actually. I've seen that cause problems in multidisciplinary care. So, it's a barrier. Yeah, poor relationships between clinicians in a multidisciplinary team is a barrier, and that has a lot to do with personality" (i32, S)<br><br>"I think everyone knows that [CPGs are] there and they should use [them]. It's probably just a case of having some people that are going to be obstinate, and just do their own thing. That's often just their [professional] culture" (i21, RO)<br><br>"Yes, I think the people that aren't adhering, are people that have been practicing for a longer period of time, and they've been using a treatment that they are used to and comfortable with, and within their own clinical practice they believe, I'm not sure if that's true, but they believe that their outcomes are good with that approach, and therefore don't, aren't interested in changing the way that they treat patients based on a guideline that doesn't fit with their own clinical experience and level of comfort" (i28, RO)<br><br>"Strict adherence to [CPGs] stops common sense from prevailing, when you're working out doses of chemotherapy for example, and other drugs. I won't go into the details but, well simplistically, people are happy, if you have too many side effects, you reduce the dose, but if you're having absolutely no side effects, but you're sticking to the protocol dose, so everyone says 'I'm good, I'm happy with that', where in actual fact you're probably underdosing the person. So that's the major issue, that causes the big variation…I'm not a big fan of, just blind adherence to guidelines. I think that can be counter intuitive and prevent academic development within a particular field, cookbook medicine. They just pick it up and they think they're doing the right thing but it's not really, they're not" (i7, MO)<br><br>**Facilitators**<br>"We have our own department haematology protocol manual, which we will use often in on-call situations and things. So, it can mean we don't have to ask for advice as often as usual and we have a standardized approach" (i16, H) |

*(Continued)*

**Table 3.** (Continued)

| Themes/subthemes | Quotes |
|---|---|
| SUBTHEME 2.2: Generational and disciplinary differences in perceptions towards CPGs | **Barriers**<br>*"There are some senior, more senior oncologists who may stray away from those particular protocols, that's, like I said, they understand the data quite well, they have seen it evolve and they understand what the differences are. That being said, there are some senior oncologists who do do dodgy things, but not in this practice, or not in the practices I'm involved in, so protocols are important"(i3, MO)*<br><br>*"80% of prostate surgery for instance, happens in the private system. There is an in-built bias against men actually going to see a radiation oncologist, because it could take work away from surgeons" (i31, RO)*<br><br>*"If you're a surgeon who relies on, entirely on private practice then you would be talking yourself into doing operations so that you don't lose that operation…when it's clearly for the patient's benefit, the best thing is to have chemotherapy first, but a surgeon operating in private, wants to do the operation" (i14, S)* |
| SUBTHEME 2.3: Litigation concerns | **Barriers**<br>*"Deviation from dosing has been very topical and I think that can be both good and bad. I mean, I think perhaps in the past, one might have tended to look at one's patient and said, 'Well maybe we need to back it off a little bit', and perhaps the recent litigation has made one less inclined. So, you could argue that that's maybe not necessarily a good thing for patients" (i11, H)*<br><br>**Facilitators**<br>*"Guidelines would help guide, particularly cases that are sort of borderline ones, say, when you are really not quite sure… Guidelines tend to be based on evidence and best practices, peer acceptance of data, all that stuff. So, that's part of the reason why we adhere to them because it's peer accepted so to speak for. And medico legally it's sound as well, you know, it's something they can fall back on" (i17, S)* |
| SUBTHEME 2.4: Patient age, comorbidities, preferences and logistics | **Barriers**<br>*"Particularly in early breast cancer, there are a lot of women who choose not to follow advice. I wouldn't say a lot, but an alarming number choose to treat it with alternative therapy or different lifestyle choices and that's just part of life, a part of medicine and part of humanity. And it's perfectly their right to do that, so long as they've been fully and properly informed" (i5, MO)*<br><br>*"When you have patients where definitely, [they] have understandably, have really no concept, no idea what treatment entails. And believe me, a lot of them might be from different cultural backgrounds, where their concept of chemotherapy already is very different from our concept of chemotherapy. Where they imagine a quite actually ghastly process of chemotherapy…it is very difficult to try to convince a 70-year old patient that we have gentle chemotherapy for them, that they won't lose their hair or have vomiting, they don't believe us, understandably…. So that's a huge barrier because you kind of need to then talk them through these concepts, already there is a language cultural barrier. So, it's very hard to navigate through and unfortunately, we have to, certainly for some of my patients from different ethnic backgrounds, where you end up giving them less intense chemo than what we have recommended, just out of sheer resistance and fear to the point where they would say 'I'd rather not get chemo'" (i10, H)*<br><br>*"You might have an elderly patient that with lots of comorbidities, and I would think, they probably wouldn't get through that treatment, so even though the guideline says that this patient fits into this treatment, but in reality, they would probably struggle with that treatment. They might actually die on treatment rather than actually being able to finish treatment. So, that's a conversation that I would have with the patient and if the patient chose not to be treated on that guideline, I would totally agree with them" (i2, RO)*<br><br>*"There's a tendency not to refer elderly patients for chemotherapy. When they've got good function, there's absolutely no reason why they shouldn't go and get an opinion" (i14, S)*<br><br>*"Age, I think, is often very inappropriately used rather than a person's fitness. And there's a lot of comorbidity assessment that's very dodgy and especially set up to fit your prejudices" (i27, RO)*<br><br>*"The particular circumstances of your patient, for example, in radiotherapy treatment, geography plays a big part in how far you live from the centre, and how many times someone can come back and forth from wherever they live. Sometimes you'll adjust the actual treatment accordingly to actually allow the patient to be able to attend the consultations. You could find, in the literature, other ways of treating the patient that might get away with less treatments or less often, but it might not necessarily make the guidelines" (i12, RO)* |
| **Theme 3 Access to, awareness of and availability of CPGs** | |

*(Continued)*

**Table 3.** (Continued)

| Themes/subthemes | Quotes |
|---|---|
| SUBTHEME 3.1: Access to, awareness of and availability of CPGs | **Barriers**<br><br>"Everybody says they [adhere to CPGs]. Everybody thinks that they do it, whether they do it or not is another matter" (i27, RO)<br><br>"That's always a slight barrier to have to login [to the CPG website]. Once you forget the password, you kind of give up" (i21, RO)<br><br>"As a hematologist we are somewhat different to the majority of medical oncologists, because a lot of what we do, there are certainly no Australian guidelines. There are overseas guidelines, and occasionally articles published in…journals. There are protocols on eviQ and that's about the extent of a lot of what we do, mainly because hematology is, all of our diseases are really rare in comparison, to most of what the oncologists treat. So, there isn't always consensus about the best way to treat everything" (i4, H)<br><br>**Facilitators**<br><br>"If they're widely distributed and easily accessible" (i30, S)<br><br>"I think apps would be good. Having them on the phone. It's just that even if there was an app and it was beautiful, I wouldn't be able to use it here because there's no Wi-Fi. I can't get phone calls here. So, the hospitals need to have inbuilt support 'for tech'" (i20, MO)<br><br>"We have access to everything about the guidelines. Personally, I have them already in my pocket [on my phone]" (i13, S)<br><br>"Accessibility is obviously a good thing. EviQ you used to have a password which they don't have anymore…It was only a minor barrier, but I think it removes it because I could never remember my password, so personally I found that quite good" (i25, H) |
| **Theme 4: Organisational and Cultural factors** | |
| SUBTHEME 4.1: Access to treatments recommended by CPGs, resource availability and clinician time | **Barriers**<br><br>"A lot of what we do is driven by what the PBS allows us to do. So, if we take the example of myeloma, there's lots of various treatment options overseas that use a number of potential combinations of novel drugs. They are quite expensive in Australia. In the Australian environment, the choice of what to prescribe really comes down to what's available on the PBS. And that's one of 2 different regimes. So that as much as anything drives how we treat these, a lot of diseases, it's a case of what we have access to …so the guidelines that there are, often overseas ones, and they are written from a, not from the Australian context, so they might recommend a whole lot of treatments that are not readily accessible. Really the only Australian guidelines there are, are sort of the brief notes in eviQ protocols, but that's not really guidelines" (i4, H)<br><br>"A major issue, a classic example is that the ASCO guidelines recommend that the drug [drug name] be given to every post-menopausal woman who has had adjuvant treatment for high-risk early breast cancer. The drugs are not available or even approved for that indication in Australia. These are real difficulties, and the system lags a long time behind that information coming out there. Usually TGA is fairly rapid, but the PBS process of course is rightfully so, very conservative. They even want cost effectiveness data, not just raw survival improvement" (i5, MO)<br><br>"In Australia… it's a PBS restriction. So it may be that you know if, for example, we have new fantastic treatments for young patients with CLL, but the guidelines in Australia are updated in accordance with what the PBS allows us to prescribe. So, the guidelines are relevant to what we can do. Even if the evidence is ahead of the guidelines, but the evidence is also ahead of what we can prescribe" (i25, H)<br><br>**Facilitators**<br><br>"Organisational support or having a culture of the team adhering to certain guidelines, so look, I guess a lot of this is communication. Communication in so many ways. What would help me to adhere to guidelines more? Accessibility. It's knowing that peers follow certain guidelines, so if I had peers saying I follow these guidelines, I'd be inclined to do that too. If we had our clinical leaders suggesting guidelines, then I would follow those" (i2, RO)<br><br>"There's always a delay [between evidence and PBS funding] and it's always too long, as far as a practicing clinician is concerned. But you know we live in the system that we do and thankfully companies will provide access [to drugs] often" (i15, MO)<br><br>"The advantages of [CPGs] are they give you, like I said, confidence in your decisions, added confidence in decisions that you're making are clinically safe and medico legally backed up, I guess. For want of a better phrase, and guidelines, they summarise the evidence, really well, so they save you time" (i28, RO)<br><br>"For people to have protected time…So for my point five clinical, I have three clinic, one theatre and contouring. So, guidelines or reading…for me is Saturday, Sunday, after hours. Yeah. And I presume it's the same for my colleagues. So, if you try to do everything within hours, it's impossible. Yeah. So, it is hard. But, I think if you give people or acknowledge the time that they read guidelines and read articles, as part of the core business that would encourage people to do that" (i23, RO) |

*(Continued)*

**Table 3.** (Continued)

| Themes/subthemes | Quotes |
|---|---|
| SUBTHEME 4.2: A culture of peer or multidisciplinary review of treatment plans | **Barriers**<br><br>"One of the big issues that's a very unspoken thing, I think in haematology, that there are some clinicians who don't work in a big unit, where they're isolated, on their own and in private practice, and maybe in smaller more rural areas potentially, where they don't have the benefits of an MDT. Where they're not able to access, really you know, the ability to go to conferences because they're not covered by their colleagues. You will get, and we've certainly seen cases, where you going to have clinicians who are not up-to-date, who you know if I speak frankly, they're not providing probably optimal care to some of the cancer patients, and those are the ones where I think standardised care model will be very, very helpful for them to look up" (i10, H)<br><br>"For those who don't attend MDT's, so surgeons who work only in private, surgeons who are a fair distance away, if you work in [rural area] for example which is about 60 minutes from [hospital Y], the guys in [rural area] don't have their own MDT. I mean they are working at public hospitals, they operate on breast cancers, but they don't, their cancers are presented at MDTs, but they don't attend our MDT. So, I bet you that the bloke down there doesn't, probably isn't aware of the current guidelines, I mean I know that he probably isn't aware, because he's doing a lot of things out of guidelines. . . Once you wander out from the major hospitals where there's less peer examination, like the guy in [rural hospital], he doesn't stick to any of the guidelines" (i17, S)<br><br>"You would have people who as we previously alluded to, depending on their craft group, they're used to making individual decisions and therefore don't tend to use guidelines as much. They have to base on their experience because that's what guides them through. And also, I guess, the way they were trained. If you were an oncologist or a surgeon who trained in or who were brought up in a group that totally adhere to guidelines, when they practice, they would do that. So, I think that upbringing or that training period is essential. . .and I think remote practice doesn't have peer review, doesn't have the critical mass of the group, and doesn't have access to MDT maybe that's a barrier" (i23, RO)<br><br>"Only about 13 percent of men who have prostatectomy, had a consultation with a radiation oncologist beforehand. So that's I think clear evidence people aren't being exposed in a multidisciplinary environment" (i27, RO)<br><br>"That's where the discussions in the MDT become heated. And the Americans will give every T3 cancer radiotherapy whether it's this early T3 or not. In Europe they would be very selective about whether the T3 cancer needs radiotherapy or not. So, there's the side effects of radiotherapy are significant and that's a battle in the MDT. So, I've attended [numerous] MDTs in Sydney on a relatively frequent basis and so [Hospital A] is very aggressive, [Hospital B]'s in the middle, [Hospital C] is, should give more radiotherapy than they do and [Hospital D] has it right" (i14, S)<br><br>"Personal interaction is a way that we all keep up-to-date with other specialties, and the roles that might pertain to those different specialties. So, if you're a radiation oncologist for instance, and you think that everyone with metastatic melanoma should have brain radiotherapy for brain metastases, you know, and you don't talk to colleagues who are using immunotherapy for instance, then you just won't ever change because you'll always be thinking I'm holding on, you know, everything looks like a nail if you're holding a hammer" (i15, MO)<br><br>**Facilitators**<br><br>"I think what the concept of multidisciplinary teams does, is to get people who are otherwise not engaged in cross-disciplinary discussion, to engage in those discussions and to refer patients that they might otherwise not refer. And so, the MDT, in actual fact, probably isn't so much about ensuring that guidelines get done. It's about ensuring that patients do have multidisciplinary care. And so, in that sense I think that that's the desire for people to be discussed, and multidisciplinary meetings is about trying to ensure that a surgeon who thinks that somebody just needs surgery, gets more than just surgery. And you know it, that's probably the greatest benefit for all of the MDTs as far as I'm concerned. . . multidisciplinary care works when people get engaged in the process. The application of guidelines obviously is part of that process, but it's not the entire kind of recipe for success" (i15, MO)<br><br>"If you're within a multidisciplinary team setting. . . you're more exposed to current accepted practices and testing. And there's also the peer, there's always some degree of peer acceptance as well. If you're doing something completely left field, you'll just look like an idiot" (i17, S) |

*(Continued)*

**Table 3.** (Continued)

| Themes/subthemes | Quotes |
|---|---|
| SUBTHEME 4.3: Referral pathways | **Barriers**<br><br>"The oncologists bring patients to the MDT who are, who have been planned to have surgery where it's highly inappropriate. This is by other surgeons. The surgeon decides to do an operation before radiotherapy. No one would defend that decision. And that's what we do find occasionally in the MDTs. Oncologists bring these patients up and you think 'Oh Christ, really? And unfortunately, I've been forced into ringing these surgeons, and they get very stroppy… This is the barrow of personality. There's a strong radiotherapist personality in (Hospital X) who is very anxious about pelvic sidewall nodes and overly so, and wants everyone irradiated, so if a patient with rectal cancer gets referred to a radiotherapist rather than a surgeon, it's quite hard to stop the radiotherapist from doing radiotherapy first" (i14, S)<br><br>"In Australia, only about 8% of men who go on and have a radical prostatectomy, the surgical removal, have actually seen a rad onc [radiation oncologist] beforehand. And that's because, yeah, this is one of the big problems, they'll link to guidelines, but the fact is, the problem is, the surgeon is always the one that makes the diagnosis. So, all men with prostate cancer would have seen a surgeon, but, only if men get referred by the surgeon or by their GP or by patient enquiry, will they get to see a radiation oncologist. So, there is already a massive bias against men actually getting all these opinions so they can actually make an informed decision" (i31, RO)<br><br>"In terms of the referral patterns as radiation oncologist, GP referrals direct sort of referrals from their primary carer, are relatively uncommon. So that's not usually the way that we get our patients. So occasionally you know, GPs will refer directly, but typically they'll go via the surgeon who did the biopsy or what have you. So, the GP will find a lump, send them off to see the surgeon, the surgeon does the biopsy, or does the surgery and then sends them to us. And so yes that's kind of the way that the referral patterns typically work. So, we might not have a choice, and you know you sort of have to deal with what you're given in terms of what treatment the patients already started on, in terms of their pathway" (i18, RO)<br><br>"Part of the diagnostic pathway. So, the first port of call is who does that sort of first diagnosis, and for a lot of tumors, it is a surgeon because it's a procedure. Some exceptions to that would be lung cancer where it's a respiratory physician who might be the first referral, who organises the biopsy and then refers on. But, yes, I do think the fact that, because the surgeon is the first port of call sometimes, that leads to a lack of multidisciplinary engagement and patients proceeding down a path that is in line with the surgical, the surgeon's preferences, which may or may not be guideline concordant or concordant with patient preference" (i29, RO) |
| **Theme 5: CPG Development and implementation factors** | |
| SUBTHEME 5.1: Development, adaptations and review of CPGs, by an expert development committee | **Barriers**<br><br>"Well, I think eviQ clinician inputs are good. I think getting people onto the committees is difficult. It's voluntary labour and people are busy, and I think that tends to be a case as in all things 'a willing horse is often saddled'. So, you end up with a small group of energetic people who are prepared to make the guidelines" (i11, H)<br><br>"Sometimes you see guidelines come out and there might be, you know, 10 surgeons and one medical oncologist and one radiation oncologist. And so that might not be appropriate representation of all the specialties across guideline development, which I guess, can make people a bit more sceptical about the guidelines" (i31, RO)<br><br>"They're all American. American medicine is also impacted by commercial interests a lot more than what Australian or English or European medicine is. So, [the CPGs are] very surgically quite heavy handed. And it's much more heavy handed than what we would be. So, they may operate on something which I think is palliative. I can't say with absolute certainty what drives that, but surgeries are a good earner. These institutions in the United States are dependent on insurance money to maintain it. So, you know, that may influence them, whether they know it or not" (i32, S)<br><br>**Facilitators**<br><br>"In uro-oncology, a lot of people reference the EAU guidelines which is the European Association of Urology, and that has quite a strong underlying urological surgical bias, from my point of view. Whereas NICE guidelines, I feel are quite balanced, because they have very good involvement of Clinical Oncology, surgery, nursing representation and patient representation right. So, the constituency of the guidelines does influence how I perceive it. In terms of, is it an adequate multidisciplinary representation" (i29, RO)<br><br>"The evidence base of course, and the methodology which they have made the guidelines, the sort of people who have reviewed the guidelines who are making the guidelines, and also, I guess the various professional bodies who have endorsed the guidelines. Trust in those organisations… you really want the experts to review… we want the people in the various professional bodies with enough gravitas, that come from a real you know, real expertise. That's what you want" (i10, H) |
| SUBTHEME 5.2: CPG Dissemination and Implementation Strategies | **Barriers:**<br><br>"(Clinical audits don't) reflect the ground reality. Like, if we have deviated from the guidelines for a particular reason, so if you do a quality control on all our patients for example, may see a 30 percent deviation because those guidelines are not suitable for those particular patients" (i22, H)<br><br>**Facilitators**<br><br>"Endorsement by peak bodies or leading clinicians in this space always helps, and also knowing, having the ability or transparency about who is part of the guidelines. You know there are people that you are aware are involved in research, have a high level of expertise, you know, all of those, makes me more likely to want to use it, follow it, trust it" (i28, RO) |

(Continued)

**Table 3.** (Continued)

| Themes/subthemes | Quotes |
|---|---|
| SUBTHEME 5.3: Suggested development and implementation improvements | "I think those international collaborations a very important, but I think to get more of that granular detail at where you need to get more country specific, to say well these are the international guidelines, but this is what will work in a certain country given our technology and our funding model. So, I think being able to use that and then adjust it" (i2, RO) |
| | "It would be nice if there was a national guidelines consortium that was properly resourced and kept current that would be great. . . it would be good to have some central coordinating credible committee or credible organising group" (i27, RO) |
| | "If the guidelines were a little bit more of a living document, you know, like a wiki that could be updated. I'm very impressed by the head and neck guidelines, which have moved to a wiki format. That seems to solve a lot of the problems that they've had maintaining currency" (i33, S) |
| | "Trainees and emerging doctors are looking for other ways to add to their brand and to be employable and an attractive candidate to work in institutions. So, it would appeal to junior doctors. . . But that's another point like because they are the ones who are working with different clinicians. They're the ones doing all the work, and seeing all the new patients and they are learning so they're fresh, and if they don't have the experience but it would be great to see what barriers they're seeing in daily practice and give them a little bit of an edge to their CVs. But yes, you do need levels of experience as well. You know people who have seen what's worked and what hasn't" (i20, MO) |
| | "Maybe there could be some incentives, I mean, a lot of what we do is, people do as volunteers. And I'm not saying that you have to be paid for everything. But, you know, if you work in the private and you're giving up a session. And it does happen with the same general practitioners. Then actually maybe some financial incentives could be there, professional development incentives, yes, maybe some kind of recognition if the publication is going to come out of it. People want authorship to improve their CV. People might be more willing to do it" (i31, RO) |
| | "EviQ has been very successful model and one would hope for the future that it expands its remit taking treatment algorithms rather than just the recipes I mentioned before. . . Look I think the eviQ framework is very good . . . it's web based. It's easy to access and I think what really needs is in addition to the actual protocols of some algorithms in terms of sequencing"(i11, H) |
| | "A sequence of things. NCCN does that well. They do flow charts. That's a good idea. I'm a picture person, and I think a lot of patients understand that" (i20, MO) |
| | "People sometimes are dismissive of real-world data, because it's not as clean as clinical trial data but the reality is, a lot of the patients we treat wouldn't meet the criteria to be enrolled in clinical trials, and yeah that's the major use of real-world data, is to try and somehow tease out some ideas. Perhaps not the majority of patients but, a significant proportion of patients would not be eligible for trials to support the treatment that we're using. . . but there is a big push to try and broaden trial criteria, if possible"(i4, H) |
| | "Printable formats for patients are always good. Different languages. . . anticipating what's coming next is very important for the patient experience, and guidelines can help, you know, do that if they're presented in a really sequential format" (i20, MO) |

considered barriers to adherence. CPGs often have clear treatment options for first-line treatment but were criticised for having less clarity for second and third-line treatment options due to a lack of evidence, leading to practice variation.

## Theme 2: Individual clinician and patient factors

### Subtheme 2.1: Clinician personality, and the impact of CPGs on autonomy

**Barriers:** Multiple clinicians highlighted that CPGs are guides, or frameworks, that support decision making, but require clinicians to apply clinical judgement when making clinical decisions, reinforcing that CPG recommendations should not be considered rules to which clinicians should strictly adhere.

Clinicians reported that the personalities or hubris of influential clinicians can act as barriers to adherence, with strong personalities influencing treatment decisions in MDTs. Clinicians suggested that individual clinical equipoise can impede clinician acceptance of new evidence-based treatment options, and many noted that as subject experts they no longer needed to regularly refer to CPGs.

**Facilitators:** A positive sentiment captured by clinicians (including a registrar), was that CPGs enable junior clinicians to have more autonomy, as it provides them with an independent mechanism to confirm treatment plans.

### Subtheme 2.2: Generational and disciplinary differences in perceptions towards CPGs

**Barriers:** Generational differences in clinician attitudes and use of CPGs was raised, with CPGs being considered less helpful for experienced clinicians, who may be less inclined to refer to CPGs, compared to junior clinicians. Junior clinicians' practice was also perceived to be influenced by the preferences of senior clinicians, potentially acting as a barrier to adherence.

Clinicians also raised concerns that clinicians can be biased toward their own discipline, or financially incentivised by fee-for-service, to independently complete treatment with patients rather than engage in CPG-adherent multidisciplinary care.

### Subtheme 2.3: Litigation concerns

**Barriers:** Clinicians raised concerns that following guidelines, due to apprehension about litigation for non-adherent practice, could lead to patients not receiving the best practice.

**Facilitators.** Possible litigation (although rare) was a strong incentive for clinicians to adhere to CPGs, encouraging clinicians to justify and communicate treatment decisions clearly, and providing assurance and medicolegal protection that clinicians are practicing according to the evidence.

### Subtheme 2.4: Patient age, comorbidities, preferences and logistics

**Barriers.** Clinician concern about patients' older age, frailty, fitness, performance status, comorbidities, contraindications, and organ impairment, can all act as barriers towards CPG adherence. Clinician concern about toxicity or potential side effects of a treatment were also seen as barriers, including concern about how patient treatment history may affect future treatment tolerability (for example, past radiation on present treatment plans).

Similarly, patient preference, and concern about side effects, toxicity, and treatment tolerability can also impede receipt of CPG adherent care, with some patients rejecting treatment plans based on anecdotal experiences of friends and family receiving treatments. Geographic

challenges and the logistics of patients travelling long distances to access treatments also contributes to lower CPG adherence, necessitating alterations to treatment schedules.

## Theme 3: Awareness of, access to and availability of CPGs

### Barriers

Clinicians commented that CPGs can be hard to access, especially if published in a journal that is not open access, or on a website that requires clinicians to login (as passwords are often forgotten). Poor Wi-Fi access and internet site restrictions in hospitals can limit real time access to CPGs. Several clinicians indicated that there weren't many (if any) local Australian CPGs available in their field, particularly for rare cancers while often international CPGs were not applicable locally. Clinicians observed that other clinicians' limited awareness of CPGs or limited knowledge of where to access them acted as barriers to adherence.

### Facilitators

Clinicians felt that easy access to guidelines facilitated use and adherence. CPGs that were available electronically or via phone applications (apps) were easier to access, as were those published in open access journals or published in a peer-reviewed, reputable journal, and free to download. Availability of CPGs on websites or apps that required no password was considered a facilitator. Some clinicians expressed a preference for local CPGs, or protocols produced by their hospital departments. Others preferred international CPGs, as they tend to be more frequently updated. All clinicians said they were aware of CPGs in their field, an important facilitator of adherence. It is important, however, to remain cognisant that CPG awareness doesn't necessarily translate to adherence.

## Theme 4: Organisational and cultural factors

### Subtheme 4.1: Access to treatments recommended by CPGs, resource availability and clinician time

**Barriers.** Limited access to resources such as drugs and technology impacts adherence. In Australia, this occurs when international CPGs recommend drugs that are not approved by the Therapeutic Goods Administration (TGA) or funded by the Pharmaceutical Benefits Scheme (PBS) (Australia's approval authorities), limiting their availability, and increasing costs. In these situations, clinicians weigh up the cost-benefit of international CPG-adherent treatments. High clinician workload, limited staffing, and lack of clinician time were also regarded as barriers to CPG adherence.

### Facilitators

Clinicians explained that when a CPG-recommended drug is not approved or funded in Australia, some access schemes operated by pharmaceutical companies or Local Health Districts, enable patients to receive those drugs. Organisational support and provision of adequate resources were seen to facilitate CPG adherence. The availability of care coordination for scans and treatment, as well as the infrastructure and use of flexible home-based treatment for geographically isolated patients were also seen as facilitating factors.

CPGs were perceived to save clinicians' time by concisely summarising the evidence and were considered a better alternative than clinicians searching through the literature independently, so long as they are up-to-date. Clinicians suggested that regular meetings to discuss

CPGs, protocols, and practices, and the purposeful hospital provision of protected time for clinicians to read, discuss and contribute to CPGs and the literature encouraged CPG adherence.

### Subtheme 4.2: A culture of peer or multidisciplinary review of treatment plans

**Barriers.**   Limited access to peer review or multidisciplinary review of treatment plans for private practicing clinicians and rural and regionally practicing clinicians, and poor multidisciplinary engagement or poor MDT attendance, were seen to contribute to lower CPG adherence. Clinicians noted that peer review occurs less frequently for common cancers.

**Facilitators.**   Multidisciplinary engagement or MDT attendance, and a culture of valuing multidisciplinary care was seen to facilitate CPG adherence reinforcing how important peer review of treatment decisions was. CPG-focused clinician training was seen to produce clinicians more inclined to adhere to CPGs.

Clinical leadership that encourages CPG adherence, a culture of error reporting, and documenting treatment decisions facilitate adherence. Several clinicians commented that peer expectation to adhere to CPGs was an influential factor, as was fear of looking negligent if non-adherent. Good relationships between multidisciplinary teams, teamwork and timely peer support were also seen as important facilitating factors.

### Subtheme 4.3: Referral pathways

**Barriers.**   Incomplete patient referral pathways were flagged as a potential barrier to CPG adherent care, particularly if patients receive treatment (such as surgery) prior to MDT presentation, potentially preventing multi-modality GRT from being delivered in the recommended sequence. Similarly, a lack of awareness by GPs (and patients) of the importance of multidisciplinary review was considered a barrier, as it can limit referrals to multidisciplinary clinicians.

## Theme 5: Development and implementation factors

### Subtheme 5.1: Development, adaptations, and review of CPGs by an expert development committee

**Barriers.**   When CPGs are perceived to be biased toward a particular modality of treatment, by development committee or individual member agendas (with biased weighting of evidence), or by pharmaceutical company influence on the development committee, this was a barrier to adherence. Clinicians also acknowledged that the development, updating, and maintenance of CPGs was seen as a slow and difficult process.

**Facilitators.**   It was seen as important by clinicians that CPGs were developed by trusted and respected experts in a transparent and methodical way, with multidisciplinary and patient representation on the development committee to avoid bias.

### Subtheme 5.2: CPG dissemination and implementation strategies

**Barriers.**   Several clinicians felt that audits of adherence rates do not accurately reflect the reasons for modifying CPGs, or take individual patient needs into account, highlighting that low CPG adherence may reflect a poor-quality CPG.

**Facilitators.**   Endorsement of CPGs, and education sessions provided by trusted and well-known organisations such as tumour groups were seen to increase clinician awareness and adherence. Similarly, effective marketing and distribution, publication in high quality journals, and discussion at conferences increase awareness and facilitate adherence. Several clinicians

commented that clinical audits, and incorporation of CPGs into point-of-care electronic decision tools nudge clinicians towards adhering to CPGs.

### Subtheme 5.3: Future CPG development and implementation improvements

CPG development should involve broader clinician input, with wider consultation outside of the working group. Junior clinician involvement in the development process was suggested with the incentive of CPG authorship, as was continuing professional development points, or financial incentives.

Adapting international CPGs to local Australian needs was recommended. This could be coordinated by a nationally resourced, centralised, and trusted CPG development body with access to good infrastructure, for quick and efficient CPG development. Development of a comprehensive centralised online cancer CPG database was proposed, that incorporates a dynamic and living wiki-style process of updating provisional CPGs, an extension of the already well-respected CCA Wiki platform, and the online Australian eviQ protocol database. Clinicians could register to receive automatic alerts about CPG updates.

CPG development should incorporate more real-world data (such as registry data) to bridge gaps in CPGs where clinical trial evidence is lacking, and to support consensus-based recommendations. Clinicians suggested that future CPGs should include treatment sequencing algorithms (e.g., decision trees and flow charts).

### Frequency analysis

The frequency analysis highlighted that the most commonly reported barriers to cancer CPG adherence were when CPGs do not cater for patient complexities (25/33), were slow to be updated (23/33), or underpinned by rapidly changing evidence (19/33). Patient treatment preferences (21/33), as well as clinician concern about patients' older age, performance status, comorbidities, and contraindications (13/33), limited availability of CPG recommended drugs (19/33), and limited access to peer review or multidisciplinary review of treatment plans (15/33) were also frequently reported barriers.

The most commonly reported facilitators to cancer CPG adherence were the perspective that CPGs provide a reassuring framework for clinicians to check their treatment plans against (24/33). Multidisciplinary engagement, or MDT attendance (24/33), easy access to guidelines (19/33), and possible litigation (18/33) were commonly reported facilitators of adherence, as were transparent CPG development by trusted and respected experts (16/33), regular CPG updates (16/33), and peer review of treatment decisions (15/33). The provision of a concise summary of evidence that includes justifications and reference to the clinical trials underpinning recommendations (15/33) was also frequently reported.

Broader clinician consultation and input, with international collaboration to develop CPGs (16/33) and a nationally resourced, centralised CPG development body with access to good infrastructure (12/33) were also common recommendations for future improvements. No disciplinary trends in attitudes were identified, and the themes were present during interviews with MOs, ROs, Haematologists and Surgeons.

## Discussion

The study examined clinician attitudes towards and determinants (perceived barriers and facilitators) of cancer CPG adherence, with the intention of informing future implementation strategies for cancer CPGs. A range of barriers and facilitators to cancer CPG adherence were identified from this study, some of which appear to be unique to the Australian context when

compared to a recent systematic review of international barriers and facilitators[35]. While noting these factors, it is important to remain cognisant of the plethora of valid reasons to make warranted variations from CPG recommendations including: patient preference; the non-applicability of recommendations to complex patients; and CPGs underpinned by weak evidence or consensus.

Lack of applicability of CPGs was seen to result from CPGs not catering for patient complexities, a universal CPG adherence issue [52], as CPGs are often underpinned by evidence from clinical trials comprised of patients who are unrepresentative of real-world populations. Instead, trial cohorts are often restricted to a subset of fitter patients, with lower risk profiles, often excluding patients based on age, organ function and lack of comorbidities [53–55]. Patient age [10,11,56–58] and comorbidities [56,58–62] are factors independently associated with cancer CPG non-adherence. This observation reflects the challenges in developing CPGs, with the work-as-done (CPG adoption and utilisation) being vastly different from the work-as-imagined by the CPG development group [63]. These issues could be addressed with greater utilisation of evidence reflective of real-world patients, including observational studies [64], to guide CPG development. In addition, incentives are needed to encourage broader eligibility criteria in industry-financed randomised trials, and to promote and facilitate post-marketing trials for patient groups not covered by industry-funded trials, in part to confirm important clinical conclusions arrived at by observational research.

Guidelines are designed to standardise practice [3], and improve care [4], but the complexity of oncological treatment decisions necessitates flexibility and reflexivity by the clinician to deliver patient-centred care [65]. Cancer care is becoming increasingly more complex, translating into lengthy and multifaceted CPGs being developed, potentially influencing adherence [66]. Concern about the evidence underpinning CPGs is considerable, with a recent Australian study indicating that 18% of CPG recommendations across a variety of conditions are based on level 1 evidence, while 19% were consensus-based [67]. This links to concerns about CPGs being biased [68]. Explicit CPG declaration of committee member medical discipline and biases, and industry funding [68,69] may help to overcome these concerns.

Limited availability of Australian CPGs was discussed as a barrier to CPG adherence, specifically, when international CPGs don't apply to the Australian context. This was reported as a significant issue when CPG-recommended drugs aren't approved by the TGA or publicly funded by the PBS in Australia, restricting access to and affordability of GRTs. Prescription of off-label anticancer medication (drugs not approved by the TGA for particular clinical scenarios) is high in Australia, with up to 85% of cancer patients receiving off-label medication, many underfunded by the PBS [70].

This study identified a perceived difference in CPG adherence between clinicians practicing in rural and metropolitan areas. Rural and remote Australian cancer services face unique logistical challenges (e.g. treating remote patients who travel hours to access services), contributing to disparities and inequalities in healthcare for a quarter of the Australian population who live outside of major cities [71]. Rural cancer patients have significantly higher mortality [72,73] and a lower likelihood of receiving GRT [12,73]. The lower survival rates are attributed in part to large distances travelled by patients, delayed diagnosis and treatment times [74], and an undersupply of oncology specialists and treatment services [72,75] necessitating patients to travel to metropolitan centres for treatment [76–78]. Modification of these patients' cancer care, as a result of these geographical challenges, may impact CPG adherence. Telemedicine is one strategy that aims to reduce these disparities [79], as well as shared care between oncologists and General Practitioners (GPs) [75].

These issues are compounded by limited access to peer review or multidisciplinary collaboration for rural clinicians. Attendance or engagement with MDTs [80] and peer review of

treatment decisions increases CPG adherence [81] and is associated with improved patient survival rates [61,82]. This teamwork, along with good collegial relationships, a culture of valuing multidisciplinary care, and peer expectation to be adherent were also perceived facilitators for CPG adherence.

MDTs often facilitate referral of patients for multidisciplinary treatment [83]. Failure to refer patients to consult with clinicians from multiple disciplines, however, was a perceived barrier to CPG adherence that limits the opportunity for patients to receive multimodality GRT in the recommended sequence. GPs often refer patients to surgeons within their existing networks [84], potentially due to limited awareness of the importance of multidisciplinary review by GPs and patients. Lack of familiarity with other treatment modalities [85] and concerns about treatment side effects can also limit referrals for radiation oncology [86,87]. Treatment patterns have been found to vary widely for prostate cancer patients in Australia, for example, depending on whether patients were referred to a radiation oncologist (RO) as well as a surgeon [88] with fewer than 14% consulting with an RO prior to surgery [89]. Addressing these referral issues requires a systems-level focus, to define and promote optimal referral pathways, rather than relying on individual GPs to appropriately refer patients to multidisciplinary care, as they typically see relatively few new cancer diagnoses each year. CCA Optimal Care Pathway documents provide support for GPs to navigate the patient journey, and often recommend referral to MDTs [90], while clinicians who attend MDTs are listed on CanRefer, an online directory of oncology specialists in NSW [91]. However, more evidence is needed to understand referral patterns in Australia and associated barriers.

Perceived difference in CPG adherence between junior and senior clinicians was identified as an issue across multiple health conditions [92,93]. Differences across cancer disciplines were also discussed, with a disciplinary bias perceived to prevent some clinicians from engaging in multidisciplinary care, potentially influenced by a fee-for-service model within some Australian cancer care services. These observations highlighting clinical hierarchies and tribalism are unlikely to reflect differences between individuals, and instead represent the broader impact of the clinical culture of hospitals on clinician behaviour [94].

## Implications for research and clinical practice

Future development and implementation of cancer CPGs in Australia should utilise the facilitators of CPG adherence identified in this study. CPGs need to be frequently updated, easily accessible, provide treatment modification options, and include a concise summary of evidence with justifications referencing the evidence. Strategies should incorporate audit and feedback strategies [27,95,96], along with education-based strategies, reminders regarding CPG updates [97], and incorporation of CPG recommendations into real-time point-of-care decision support [98]. Effective implementation strategies need to consider both the CPG content and communication of that content [99].

The establishment of a centralised, trusted, and well-funded CPG development body, akin to CCA, to produced CPGs in a transparent and systematic manner is recommended. In addition, the development of an online CPG database is recommended, that provides a comprehensive range of cancer CPGs either locally developed or adapted from international CPGs. These CPGs can be frequently updated through the use of a wiki-like process, extending the existing and well-regarded CCA wiki-platform, which enables ongoing consultation, review of the literature, and automatic updates of content [33]. As clinicians report difficulties with time, it is important that protected non-clinical time be allocated to allow clinicians to participate in the crucial work of CPG development and update.

## Strengths and limitations

Strengths of this study include the use of multiple coders, member checking and triangulation of data from participants from different disciplines and hospitals across Sydney, Australia. While it is acknowledged that member checking can be perceived as a limitation, in this instance, responding participants confirmed the thematic interpretation, and provided no conflicting comments [44]. Limitations of this study include participant self-selection bias, potentially recruiting respondents who feel particularly positively or negatively towards guidelines. The characteristics of the invited non-respondents are unknown, potentially introducing bias. The member checking process was delayed due to restricted access to hospital staff as a result of COVID-19 and was conducted in March 2021. No response rate was calculated due to the snowball element of recruitment. Similarly, the sample was limited to four disciplines of clinicians who treat cancer patients, potentially excluding the views of other clinicians involved in the patient pathway, such as clinicians who provide supportive care, palliative care, or GPs who help patients navigate their cancer journey. The cohort of participants were also typically staff specialists, from SWSLHD in the first 10 years of their career, who likely work with complex cases that are typically poorly addressed by CPGs. Only one Fellow and one Registrar participated in the study, resulting in limited observations from those groups of clinicians. Only clinicians working in NSW were interviewed, and no clinicians who work exclusively in private practice or in rural centres were included.

## Conclusion

This study has identified perceived barriers and facilitators specific to cancer CPG adherence that contribute to variation from cancer-CPG recommendations across a variety of cancer streams in Australia. This research will guide the implementation of future cancer CPGs, by informing strategies that target these factors, to enhance implementation of high-quality evidence into practice.

## Supporting information

**S1 Checklist. COREQ (COnsolidated criteria for REporting Qualitative research) checklist.**
(PDF)

**S1 Appendix. Appendix A: Interview topic guide.**
(DOCX)

**S2 Appendix. Appendix B: Coding Framework.**
(DOCX)

## Author Contributions

**Conceptualization:** Mia Bierbaum, Frances Rapport, Gaston Arnolda, Geoff P. Delaney, Winston Liauw, Ian Olver, Jeffrey Braithwaite.

**Data curation:** Mia Bierbaum.

**Formal analysis:** Mia Bierbaum, Frances Rapport.

**Funding acquisition:** Mia Bierbaum.

**Investigation:** Mia Bierbaum.

**Methodology:** Mia Bierbaum, Frances Rapport, Gaston Arnolda, Geoff P. Delaney, Winston Liauw, Jeffrey Braithwaite.

**Project administration:** Mia Bierbaum.

**Resources:** Mia Bierbaum.

**Software:** Mia Bierbaum.

**Supervision:** Frances Rapport, Gaston Arnolda, Jeffrey Braithwaite.

**Validation:** Mia Bierbaum, Frances Rapport.

**Visualization:** Mia Bierbaum.

**Writing – original draft:** Mia Bierbaum.

**Writing – review & editing:** Mia Bierbaum, Frances Rapport, Gaston Arnolda, Geoff P. Delaney, Winston Liauw, Ian Olver, Jeffrey Braithwaite.

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
