## [Decision Letter · Decision Letter 0]

7 Jul 2022

PONE-D-22-08374Clinical Practice Guideline adherence in oncology: A qualitative study of insights from clinicians in AustraliaPLOS ONE

Dear Dr. Bierbaum,

Thank you for submitting your manuscript to PLOS ONE. After careful consideration, we feel that it has merit but does not fully meet PLOS ONE’s publication criteria as it currently stands. Therefore, we invite you to submit a revised version of the manuscript that addresses the points raised during the review process.

You will see from the reviewer comments that it is considered that some additional information about the design, methods and analysis will be helpful. 

We look forward to receiving your revised manuscript.

Kind regards,

Anna Ugalde, PhD

Academic Editor

PLOS ONE

Journal Requirements:

Additional Editor Comments (if provided):

Thanks for the submission to Plos One.

This is a very interesting article but as you can see from the reviewer comments there are some concerns about the method and analysis. I see that the authors have also included a content analysis as well as a thematic; is it worth revising this and focusing on the themes that are present? From the reviewer feedback, potentially the content analysis has confused presentation of the findings. Could the authors also ensure that all of the COREQ checklist are addressed? I could see many of them (e.g. gender, training of interviewer) but if this could be assured that would assist. Please ensure a clear design statement is clear as this was also identified by the reviewer.

Could you please also confirm the dates of interviewing are included.

I look forward to your response and reviewing a revised version.

Reviewers' comments:

Reviewer's Responses to Questions

**Comments to the Author**

1. Is the manuscript technically sound, and do the data support the conclusions?

Reviewer #1: No

2. Has the statistical analysis been performed appropriately and rigorously? 

Reviewer #1: N/A

3. Have the authors made all data underlying the findings in their manuscript fully available?

Reviewer #1: No

4. Is the manuscript presented in an intelligible fashion and written in standard English?

Reviewer #1: Yes

5. Review Comments to the Author

Reviewer #1: This is interesting paper, and while the findings are not surprising they remain important to disseminate.

The original purpose of EB guidelines was to provide a collation of the best available evidence with recommendations, to "guide" the clinician in making decisions regarding the patient before them. The initial intent was that the clinician would tailor recommendations/guidance to the person sitting before them, taking into account their unique situation. As the evidence base has grown to support clinical guidelines, so has evidence for poorer outcomes in some areas when recommendations in guidelines are not followed. There is also variability in the quality of guidelines.

None of these factors have been examined in the paper. The underlying premise appears to be all clinical decisions should be based on EB Guidelines, even though they are a "guide".

A major concern with the paper, while stating it complies with CREQ criteria for reporting qualitative research, is that this is not addressed. There are significant gaps in information regarding the study design which is not described at all. The findings as presented do not represent thematic analysis, and with out any description of teh research method it is hard to determine what the underpinning philosophical approach or framework was, if there was one.

The findings as presented are more in line with content analysis, which would be appropriate in this type of research. As it currently is presented there is concerns about the authors understanding of qualitative research.

6. PLOS authors have the option to publish the peer review history of their article (what does this mean?). If published, this will include your full peer review and any attached files.

Reviewer #1: No

---

## [Author Response · Author response to Decision Letter 0]

3 Aug 2022

Editor’s comments 

Thanks for the submission to Plos One.

This is a very interesting article but as you can see from the reviewer comments there are some concerns about the method and analysis. 

I see that the authors have also included a content analysis as well as a thematic; is it worth revising this and focusing on the themes that are present? From the reviewer feedback, potentially the content analysis has confused presentation of the findings. 

Could the authors also ensure that all of the COREQ checklist are addressed? I could see many of them (e.g. gender, training of interviewer) but if this could be assured that would assist. 

Please ensure a clear design statement is clear as this was also identified by the reviewer.

Could you please also confirm the dates of interviewing are included.

I look forward to your response and reviewing a revised version.

Author’s response 

Thank you for your consideration of our manuscript and the opportunity for revision.

Thank you for this feedback. 

We have removed the content analysis that was presented within the thematic analysis findings (p7-13). The summary of the frequency that codes were discussed, and the trends across clinician specialties, can now be seen at the end of the results section under the subheading Frequency Analysis, p13, and in table 2.

Please find the COREQ checklist attached.

Please find the design statement on page 4.

Interviews were conducted between October 2019-January 2020. Please see addition in text, p4

Reviewer one’s comments 

Reviewer #1: This is interesting paper, and while the findings are not surprising they remain important to disseminate. The original purpose of EB guidelines was to provide a collation of the best available evidence with recommendations, to "guide" the clinician in making decisions regarding the patient before them. The initial intent was that the clinician would tailor recommendations/guidance to the person sitting before them, taking into account their unique situation.

Author’s response: 

We agree and believe this is addressed in the manuscript. Please see below where these issues have been discussed in the manuscript: 

Discussion - paragraphs 1 and 2, p14.

Results - Theme 1: CPG content

SUBTHEME 1.1: Applicability of recommendations to patient population, p7

SUBTHEME 1.5: Prescriptiveness of CPG recommendations, p9

Theme 2: Individual clinician and patient factors 

SUBTHEME 2.4: Patient age, comorbidities, preferences and logistics, p10.

To make this even clearer, however, we have modified the introduction – paragraph 1, p3 - to briefly and explicitly include this concept as part of the framing of the research

Reviewer one’s comments 

As the evidence base has grown to support clinical guidelines, so has evidence for poorer outcomes in some areas when recommendations in guidelines are not followed. There is also variability in the quality of guidelines. None of these factors have been examined in the paper. The underlying premise appears to be all clinical decisions should be based on EB Guidelines, even though they are a guide".

Author’s response: Again, we agree. These issues are discussed in the following locations in the manuscript:

Results - Theme 1: CPG content

SUBTHEME 1.2: Degree of evidence and level of agreement with evidence underpinning CPGs, p8

SUBTHEME 1.3: Format, ease of use and references to evidence, p8

SUBTHEME 1.4: How up-to-date CPGs are, p8

Reviewer one’s comments 

A major concern with the paper, while stating it complies with CREQ criteria for reporting qualitative research, is that this is not addressed.

Author’s response: Please see COREQ checklist attached

Reviewer one’s comments 

There are significant gaps in information regarding the study design which is not described at all. The findings as presented do not represent thematic analysis, and with out any description of teh research method it is hard to determine what the underpinning philosophical approach or framework was, if there was one. The findings as presented are more in line with content analysis, which would be appropriate in this type of research. As it currently is presented there is concerns about the authors understanding of qualitative research.

Author’s response 

Thank you. We have added to the methods section to ensure the methods are clear for the reader. Please see amendments to the Methods section, p4, and p5.

See amendments in the results section to the presentation of the thematic analysis findings (p7-13) and summary of the frequency analysis (p13). Just by way of offering reassurance to the reviewer, the authorship team has published a deal of qualitative research over the years.

---

## [Decision Letter · Decision Letter 1]

1 Dec 2022

Clinical Practice Guideline adherence in oncology: A qualitative study of insights from clinicians in Australia

PONE-D-22-08374R1

Dear Dr. Bierbaum,

We’re pleased to inform you that your manuscript has been judged scientifically suitable for publication and will be formally accepted for publication once it meets all outstanding technical requirements.

Kind regards,

Anna Ugalde, PhD

Academic Editor

PLOS ONE

Additional Editor Comments (optional):

Thank you for responding to the reviewers comments. I recommend this paper is suitable for publication.

Reviewers' comments:

Reviewer's Responses to Questions

**Comments to the Author**

1. If the authors have adequately addressed your comments raised in a previous round of review and you feel that this manuscript is now acceptable for publication, you may indicate that here to bypass the “Comments to the Author” section, enter your conflict of interest statement in the “Confidential to Editor” section, and submit your "Accept" recommendation.

Reviewer #1: All comments have been addressed

2. Is the manuscript technically sound, and do the data support the conclusions?

Reviewer #1: Yes

3. Has the statistical analysis been performed appropriately and rigorously? 

Reviewer #1: N/A

4. Have the authors made all data underlying the findings in their manuscript fully available?

Reviewer #1: (No Response)

5. Is the manuscript presented in an intelligible fashion and written in standard English?

Reviewer #1: Yes

6. Review Comments to the Author

Reviewer #1: (No Response)

7. PLOS authors have the option to publish the peer review history of their article (what does this mean?). If published, this will include your full peer review and any attached files.

Reviewer #1: No

---

## [Editor Report · Acceptance letter]

8 Dec 2022

PONE-D-22-08374R1 

*Clinical Practice Guideline adherence in oncology: A qualitative study of insights from clinicians in Australia*

Dear Dr. Bierbaum:

I'm pleased to inform you that your manuscript has been deemed suitable for publication in PLOS ONE. Congratulations! Your manuscript is now with our production department. 

Kind regards, 

on behalf of

Dr. Anna Ugalde 

Academic Editor

PLOS ONE